# A pairwise cytokine code explains the organism-wide response to sepsis

Michihiro Takahama [1,2], Ashwini Patil [3], Gabriella Richey[1], Denis Cipurko[1], Katherine Johnson[1], Peter Carbonetto[4,5], Madison Plaster[1], Surya Pandey[1], Katerina Cheronis[1], Tatsuki Ueda[1], Adam Gruenbaum[1], Tadafumi Kawamoto[6], Matthew Stephens[4,7] & Nicolas Chevrier [1] ✉

Sepsis is a systemic response to infection with life-threatening consequences. Our understanding of the molecular and cellular impact of sepsis across organs remains rudimentary. Here, we characterize the pathogenesis of sepsis by measuring dynamic changes in gene expression across organs. To pinpoint molecules controlling organ states in sepsis, we compare the effects of sepsis on organ gene expression to those of 6 singles and 15 pairs of recombinant cytokines. Strikingly, we find that the pairwise effects of tumor necrosis factor plus interleukin (IL)-18, interferon-gamma or IL-1β suffice to mirror the impact of sepsis across tissues. Mechanistically, we map the cellular effects of sepsis and cytokines by computing changes in the abundance of 195 cell types across 9 organs, which we validate by whole-mouse spatial profiling. Our work decodes the cytokine cacophony in sepsis into a pairwise cytokine message capturing the gene, cell and tissue responses of the host to the disease.

Molecules, cells and tissues with immune functions are ubiquitous in the body. While the organismal nature of the immune system is vital for the host against infection, the systemic dysregulation of immune processes in response to infectious and noninfectious triggers can be harmful. For example, sepsis is a systemic host response to infection with life-threatening consequences[1]. The disease is a global health issue in need of targeted therapies addressing the short-term and long-term effects on the host[2–4]. Our knowledge of the mechanisms underlying the impact of sepsis on the body is rudimentary, as highlighted by expert consensus in the field of sepsis[5]. The timing and location of events that take place across organs other than blood during sepsis remain unclear. Sepsis is thus a clear example for which learning the multifactorial effects of the disease on the molecules, cells and tissues of the whole body is critically important for basic and clinical sciences.

A myriad of cells and molecules has been linked to sepsis. Numerous studies have established immune and endothelial cells together with cytokines and the complement and coagulation systems as key

cellular and molecular factors in the pathogenesis of sepsis[6]. However, the links between the molecular and cellular factors that produce the damaging impact of sepsis for the body have not been systematically mapped. For example, the uncontrolled, systemic activity of cytokines contributes to tissue injury and organ failure[7], but it is unclear which cytokines—alone or in combination—impact which cells and tissues across the body. This gap in knowledge is due to features of the cytokine language that make it hard to decode, such as the variations in concentrations (local and systemic), activities (pro-inflammatory, anti-inflammatory or both for any given cytokine), and interactions within a mixture of cytokines present in a tissue. As a result, we lack a unifying framework to understand how the cytokine network functions in sepsis, including the network's target cells, hierarchy, interactions and feedback loops[8].

In addition, many types of cells die or divide at abnormal rates during sepsis[6,9,10]. The number of lymphocytes drops[11,12], while that of neutrophils surges in sepsis[13], contributing to the negative effects of

[1]Pritzker School of Molecular Engineering, University of Chicago, Chicago, IL, USA. [2]Laboratory of Bioresponse Regulation, Graduate School of Pharmaceutical Sciences, Osaka University, Osaka, Japan. [3]Combinatics, Chiba, Japan. [4]Department of Human Genetics, University of Chicago, Chicago, IL, USA. [5]Research Computing Center, University of Chicago, Chicago, IL, USA. [6]School of Dental Medicine, Tsurumi University, Yokohama, Japan. [7]Department of Statistics, University of Chicago, Chicago, IL, USA. ✉e-mail: nchevrier@uchicago.edu

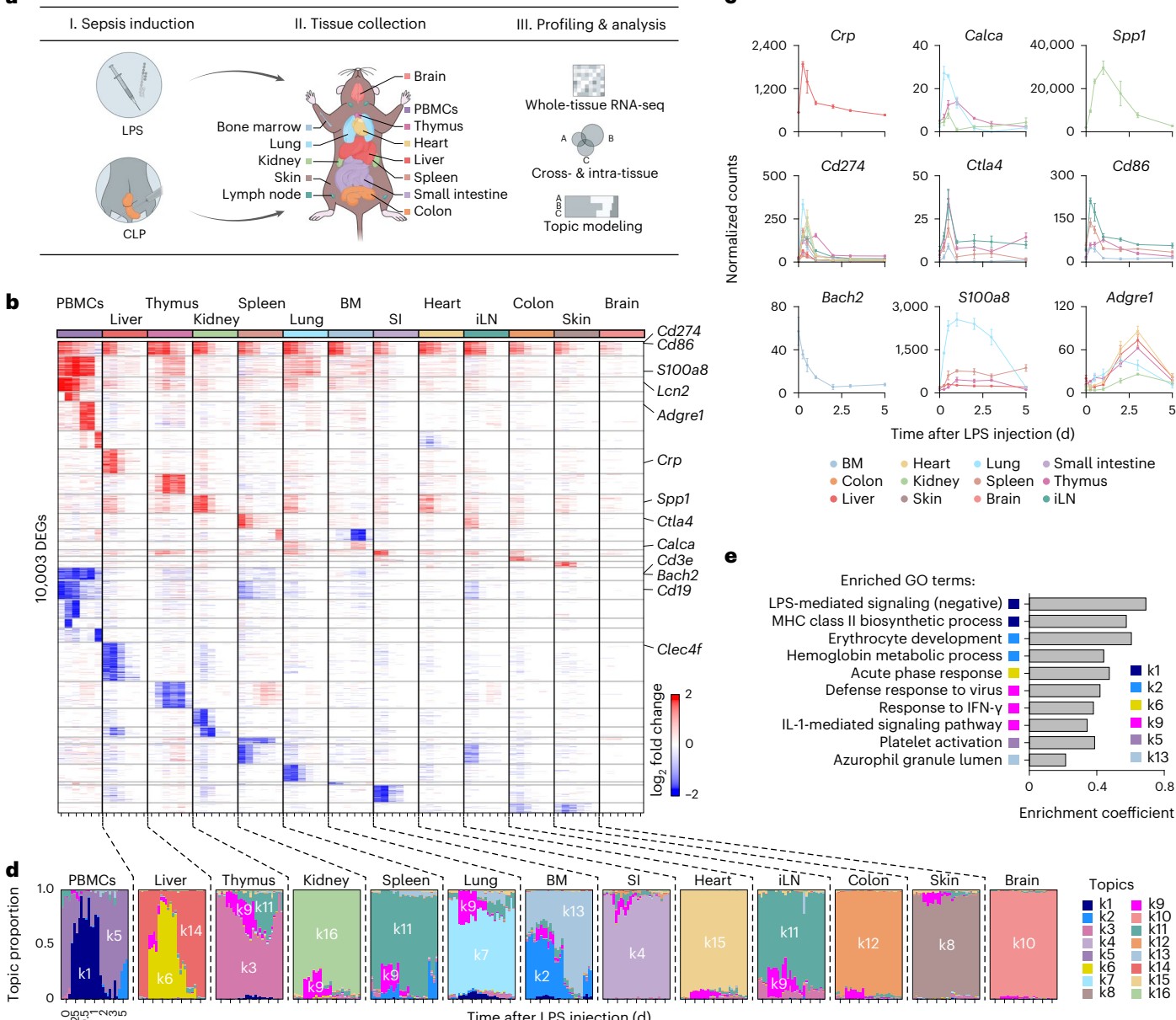

**Fig. 1 | Whole-tissue gene expression reveals the molecular effects of sepsis and endotoxemia across organs. a**, Schematic overview of the experimental workflow. **b**, Heat map of DEGs (rows) from whole-tissue mRNA profiles ordered by k-means clustering (horizontal lines), organ types (top; colors) and time periods (bottom; tick marks for 0.25, 0.5, 1, 2, 3, and 5 d) after sublethal LPS injection. Values are log2 fold changes relative to matching, untreated organ. Statistical analyses were performed with limma (false discovery rate (FDR)-adjusted P value < 0.01; absolute fold change > 2). BM, bone marrow; SI, small intestine; iLN, inguinal lymph node. **c**, Normalized counts for indicated genes, cohorts and organs (color). Error bars indicate the s.e.m. (n = 3 biologically independent samples for BM 5 d, colon 0.25 d, iLN 2 d, liver 1 d or lung 3 d; n = 4 for other groups). **d**, Structure plot of the estimated membership proportions for a topic model with k = 16 topics (colors) fit to 364 tissue samples across 13 organ types (top) from LPS-injected mice (Methods). Each vertical bar shows the cluster membership proportions for a single tissue sample ordered over time (bottom, tick marks for 0, 0.25, 0.5, 1, 2, 3 and 5 d after sublethal LPS injection) for each organ type. **e**, Pathway enrichment analysis using DEGs in each topic from **d**. Shown are enrichment coefficients (x axis) for indicated Gene Ontology (GO) terms (y axis). MHC, major histocompatibility complex.

the disease on the immune system of survivors[6,10,14]. However, we have a limited understanding of which molecules, including cytokines, are responsible for the effects of sepsis on immune and nonimmune cells across various tissue contexts[5]. Therefore, to better understand the systemic effects of sepsis, we must build a mechanistic framework explaining the causal relationships between the key molecular and cellular factors of the disease at the level of the whole organism.

Here, we used mouse models of sepsis to obtain a dynamic, organism-wide map of the pathogenesis of the disease, revealing the spatiotemporal patterns of both known and previously unrecognized effects of sepsis on the body. Strikingly, our work uncovered a hierarchical cytokine circuit arising from the pairwise effects of tumor necrosis factor (TNF) with IL-18, interferon (IFN)-γ or IL-1β, which yielded nonlinear effects on tissues through synergistic and antagonistic gene regulation. Collectively, these three cytokine pairs sufficed to recapitulate most of the host transcriptional, physiological and fitness responses to sepsis, uncovering an emerging principle in the chaotic behavior of cytokines during sepsis. Overall, our results provided fundamental

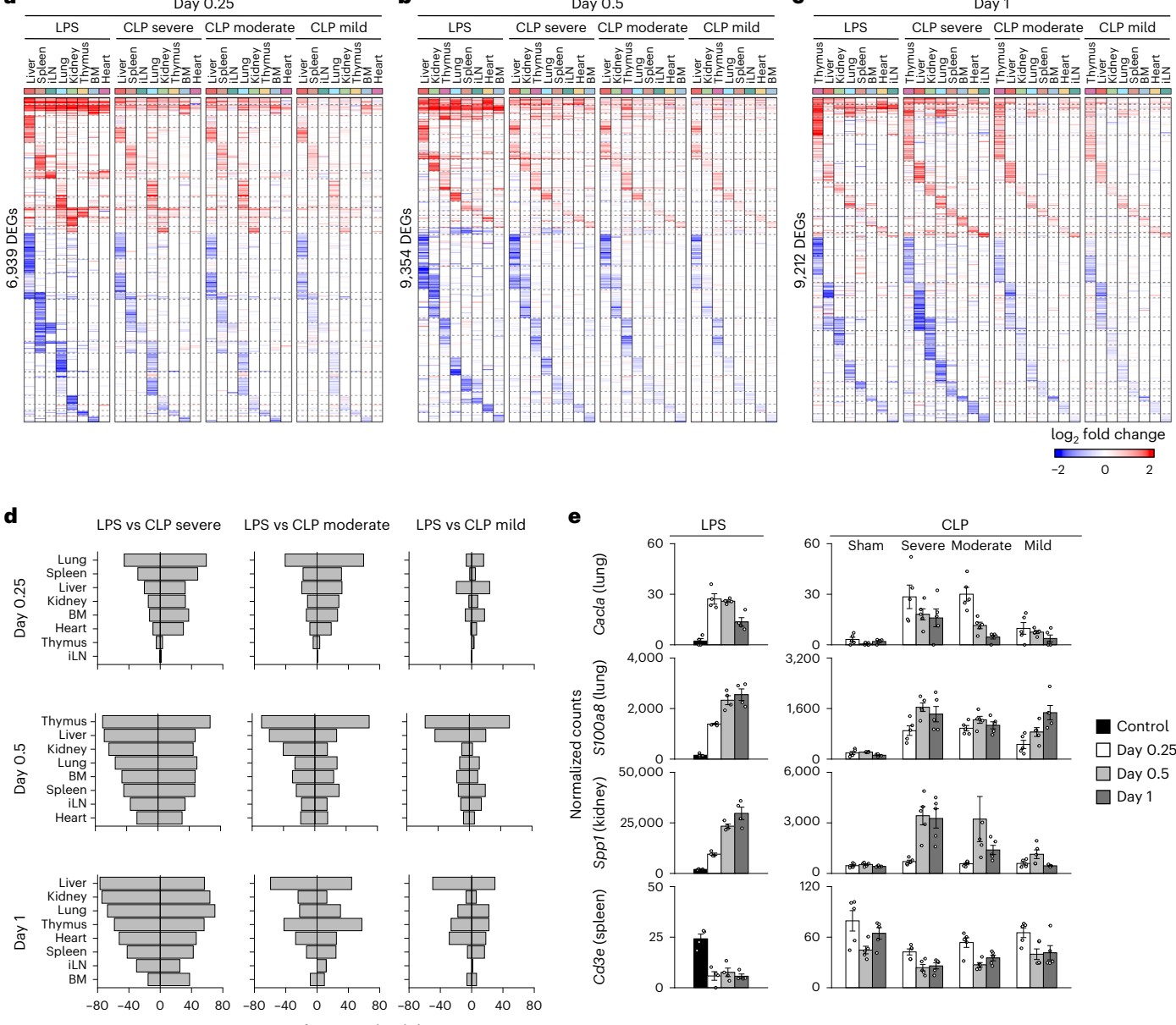

**Fig. 2 | Comparative, multi-tissue expression analysis of endotoxomia and bacterial sepsis. a–c**, Heat maps of DEGs (rows) from whole-tissue mRNA profiles ordered by *k*-means clustering and organ types (top, colors) at 0.25 (**a**), 0.5 (**b**) or 1 (**c**) day after sublethal LPS injection or severe, moderate or mild CLP surgeries. Values are log2 fold changes relative to matching organs from untreated mice for LPS or mice after sham surgeries for CLP. Statistical analyses were performed with limma (FDR-adjusted *P* value < 0.1). Shown are all the genes found to be differentially regulated in at least one of the LPS or CLP conditions for each time point. BM, bone marrow; iLN, inguinal lymph node. **d**, Percentages (*x* axis) of genes differentially expressed in tissues (rows) upon severe, moderate or mild CLP that match the genes regulated by sublethal LPS at 0.25, 0.5 and 1 d after LPS or CLP. Positive and negative percentages indicate overlaps of upregulated and downregulated genes, respectively. **e**, Normalized counts for indicated genes, cohorts and organs (*y* axes) in LPS and CLP sepsis. Error bars indicate the s.e.m. (*n* = 4 biologically independent samples for LPS samples; *n* = 5 biologically independent samples for CLP samples).

insights that will help build a unified mechanistic framework for the effects of sepsis on the body.

## Results

### The organism-wide response to experimental sepsis

To study the organism-wide response to sepsis, we measured changes in gene expression across tissues in two models leading to (1) endotoxemia using lipopolysaccharide (LPS) and (2) sepsis using cecal ligation and puncture (CLP)[15] (Fig. 1a). We profiled gene expression changes in 13 tissues[16,17], including bone marrow, brain, colon, heart, inguinal lymph nodes (iLNs), kidney, liver, lung, peripheral blood mononuclear cells (PBMCs), skin, small intestine, spleen and thymus, at 0.25, 0.5, 1, 2, 3 and 5 d after LPS injection—covering early and late effects—and from untreated control mice. In total, we identified 10,003 genes that were differentially expressed in response to LPS (Fig. 1b and Supplementary Table 1a). Interestingly, we found that nonlymphoid tissues returned to their transcriptional steady state within 5 d of LPS injection, whereas lymphoid tissues did not (Fig. 1b and Extended Data Fig. 1a), which is reminiscent of the reported link between sepsis and long-term immune defects[14,18]. At the gene level, several clinical biomarkers were upregulated such as *Crp* (liver), *Calca* (lung and kidney, early; and thymus, late) and *Spp1* (kidney; Fig. 1b,c). Costimulatory proteins Cd274

(PD-L1), Ctla4 and Cd86 linked to the immune deficiencies observed in sepsis[6,19] were upregulated across tissues (12/13 tissues for *Cd274*, 6/13 tissues for *Cd86* and 3/13 tissues for *Ctla4*; Fig. 1b,c). Cellular changes were reflected by the expression of marker genes, such as *Bach2* for erythropenia[20,21], *S100a8* for neutrophil accumulation in lungs[13], *Cd3e* and *Cd19* for T and B cell lymphopenia[22] and *Adgre1* for multi-tissue accumulation of macrophages[23] (Fig. 1b,c and Extended Data Fig. 1b). To systematically investigate how sepsis biomarkers varied in expression, we focused on 258 biomarker genes associated with sepsis in the literature[24]. We observed a range of effects including the lowest in brain with 9.7% (25/258) of biomarker genes regulated to the three highest in lung, thymus and PBMCs with 37.2% (96/258), 36.4% (94/258) and 35.7% (92/258) of biomarker genes regulated, respectively (Extended Data Fig. 1c and Supplementary Table 1b). Thus, our data reveal both intra-tissue and cross-tissue expression patterns, including genes and pathways associated with sepsis and systemic inflammation.

### Dynamic, tissue-level processes in sepsis by topic modeling

We analyzed the LPS time-series data using grade of membership models to examine the impact of sepsis on intra-tissue and cross-tissue states. Grade of membership models, also known as topic models, cluster samples by allowing each sample to have partial membership in multiple biologically distinct clusters or 'topics'[25], as opposed to traditional clustering methods that assign a sample or a gene to a single cluster. We first fit the grade of membership model to our LPS data using 16 topics, and generated structure plots of estimated membership proportions for all 364 whole-tissue RNA-sequencing (RNA-seq) profiles encompassing 13 tissues and 6 time points after LPS injection in addition to control, untreated samples (Fig. 1d). Second, to determine which genes and processes explain each topic, we used the quantitative estimates of the mean expression of each gene in each topic as provided by the grade of membership models to perform gene-set enrichment analyses (Supplementary Methods). Several topics reflected expected tissue biology such as basic functions of the small intestine (k4), lungs (k7) and heart (k15; Supplementary Table 1c). Other topics captured processes driven by LPS-induced sepsis (Fig. 1e). For example, some topics reflected an influx of granulocytes in PBMCs, which is linked to clinical deterioration[26], and, to a lesser extent, in lungs and bone marrow (k1), or to acute inflammatory response of the liver (k6; Fig. 1e). Other topics captured erythropenia in the bone marrow (k2) and neutrophil proliferation and recruitment in the spleen and lungs (k13; Fig. 1e). Lastly, topic k9 reflected organism-wide changes in interferon-stimulated genes (Fig. 1e). Topic modeling therefore delineated a dynamic view of key processes regulated by LPS across tissues.

### Similar organism-wide responses between LPS and CLP sepsis

Next, we compared the organism-wide effects of LPS to those obtained with CLP (Extended Data Fig. 2a,b), a polymicrobial infection starting in the abdominal cavity which is considered the gold standard model for sepsis due to its high clinical relevance[15]. We found a high degree of similarity between the tissue expression profiles of LPS and CLP at 0.25 d, 0.5 d and 1 d after sepsis, ranging from 29.5% in heart to 68% in thymus upon severe CLP sepsis at 0.5 d after surgery (Fig. 2a–d and Supplementary Table 2a–c). The severity of CLP correlated with the number of differentially expressed genes (DEGs) across tissues and, therefore, with the degree of overlap with LPS-induced genes (Fig. 2a–d, Extended Data Fig. 2c and Supplementary Table 2a–c). For example, genes capturing known changes in sepsis, such as the biomarker *Calca* or immune cell markers for neutrophils (*S100a8*) or T lymphocytes (*Cd3e*), followed similar changes across tissues in LPS and CLP (Fig. 2e). Taken together, our data provide an organism-wide view of the host response to LPS and CLP sepsis, including the spatiotemporal expression patterns of genes well known or previously unrecognized in sepsis (Figs. 1 and 2 and Extended Data Figs. 1 and 2).

### Pairwise cytokine effects mimic sepsis effects on tissues

To examine how much of the effects of sepsis on tissues are explained by cytokines, which are key systemic factors in sepsis and cytokine storm syndromes[6,7], we compared changes in tissue gene expression in response to sepsis and recombinant cytokines (Fig. 3a). We focused on six cytokines that play a major role in sepsis: IFN-γ, IL-1β, IL-6, IL-10, IL-18 and TNF[6]. Plasma and tissue cytokine expression patterns mostly mirrored one another and both lymphoid and nonlymphoid tissues were the source of plasma cytokines (Fig. 3b). Bimodal cytokine expression in plasma reflected different timing in cytokine mRNA induction in tissues, such as IL-10 in thymus and other tissues early on followed by spleen later, and TNF in most tissues and thymus at early and late time points, respectively (Fig. 3b). Next, we measured the effects of the six recombinant cytokines used alone or pairwise (15 pairs) on tissue gene expression (Fig. 3a). All cytokine singles and pairs led to significant changes on tissue states, ranging from 14 (IL-10) to 431 (IL-1β) DEGs across all tissues tested for singles and 12 (IL-6 + IL-10) to 7,083 (TNF + IL-18) for pairs (Extended Data Fig. 3a–c and Supplementary Table 3a–d). Strikingly, of the 6 singles and 15 pairs tested, we found a strong agreement between the genes regulated by LPS and three cytokine pairs: TNF plus IL-18 (14.9% in LNs to 56.8% in kidney), IFN-γ (3.6% in LNs to 38.2% in thymus) or IL-1β (1.9% in LNs to 28.2% in thymus; Fig. 3c,d). For comparison, the transcriptional effects of injecting naive mice with plasma from LPS-injected mice overlapped well with LPS effects (4.74% in colon to 20.5% in liver; Fig. 3c), suggesting that pairwise cytokine effects recapitulated most of the transcriptional response to sepsis. The effects of TNF plus IL-18, IFN-γ or IL-1β encompassed a high proportion of sepsis biomarker genes, 45.7% (118/258), 43.8% (113/258) or 32.6% (84/258), respectively, compared to the other 12 cytokine pairs tested (8.1% ± 7.3% s.d.; Extended Data Fig. 3d). Of the 15 cytokine pairs tested, we found that TNF plus IL-18, IFN-γ or IL-1β regulated the most genes across all organs with 7,083, 4,071 or 2,452 genes, respectively, compared to the average number of DEGs, 382 ± 298 s.d., for the other 12 pairs tested (Fig. 3d,e and Extended Data Fig. 3e).

Next, we used a linear modeling approach to classify the effects of cytokine pairs on regulated genes as synergistic, antagonistic or additive relative to their composite singles (Methods). The three

**Fig. 3 | The pairwise effects of TNF plus IL-18, IFN-γ or IL-1β recapitulate the transcriptional responses of organs to sepsis. a**, Schematic overview of the experimental workflow. Mice were intravenously injected with 6 singles, or 15 pairs of recombinant cytokines followed by RNA-seq on indicated organs. **b**, Normalized counts (top) and blood concentration (bottom) for indicated cytokine genes and proteins upon sublethal LPS injection at indicated time points. Error bars indicate the s.e.m. (*n* = 3 biologically independent samples for normalized counts in BM 5 d, colon 0.25 d, iLNs 2 d, liver 1 d or lung 3 d; *n* = 4 for other groups). BM, bone marrow; iLN, inguinal lymph node; SI, small intestine. **c**, Percentages (circle) and numbers (color scale) of genes differentially expressed upon injection with indicated recombinant cytokines (rows) across organs (columns) that match the genes regulated by sublethal LPS at 12 h after sepsis induction. 'Plasma' indicates naive mice injected with plasma from LPS-injected mice. **d**, Heat map (left) of DEGs (rows) from whole-tissue mRNA profiles ordered by *k*-means clustering and organ types (top, colors) at 12 h after sublethal LPS injection. Values are log2 fold changes relative to matching, untreated organs. Statistical analyses were performed with limma (FDR-adjusted *P* value < 0.01; absolute fold change > 2; *n* = 4). Genes upregulated and downregulated by indicated recombinant cytokine pairs in at least one of the nine tissues profiled are indicated in red and blue, respectively. **e**, Numbers of genes (*x* axis) differentially regulated by indicated cytokine pairs (rows) but not by matching single cytokines. Statistical analyses were performed with limma (FDR-adjusted *P* value < 0.01; absolute fold change > 2; *n* = 4). **f**, Percentages (*x* axis) of genes differentially expressed in tissues (rows) upon injection of the indicated three cytokine pairs that match the genes regulated by bacterial (LPS, CLP; top) or viral (WR; bottom) sepsis. WR, vaccinia virus strain Western Reserve.

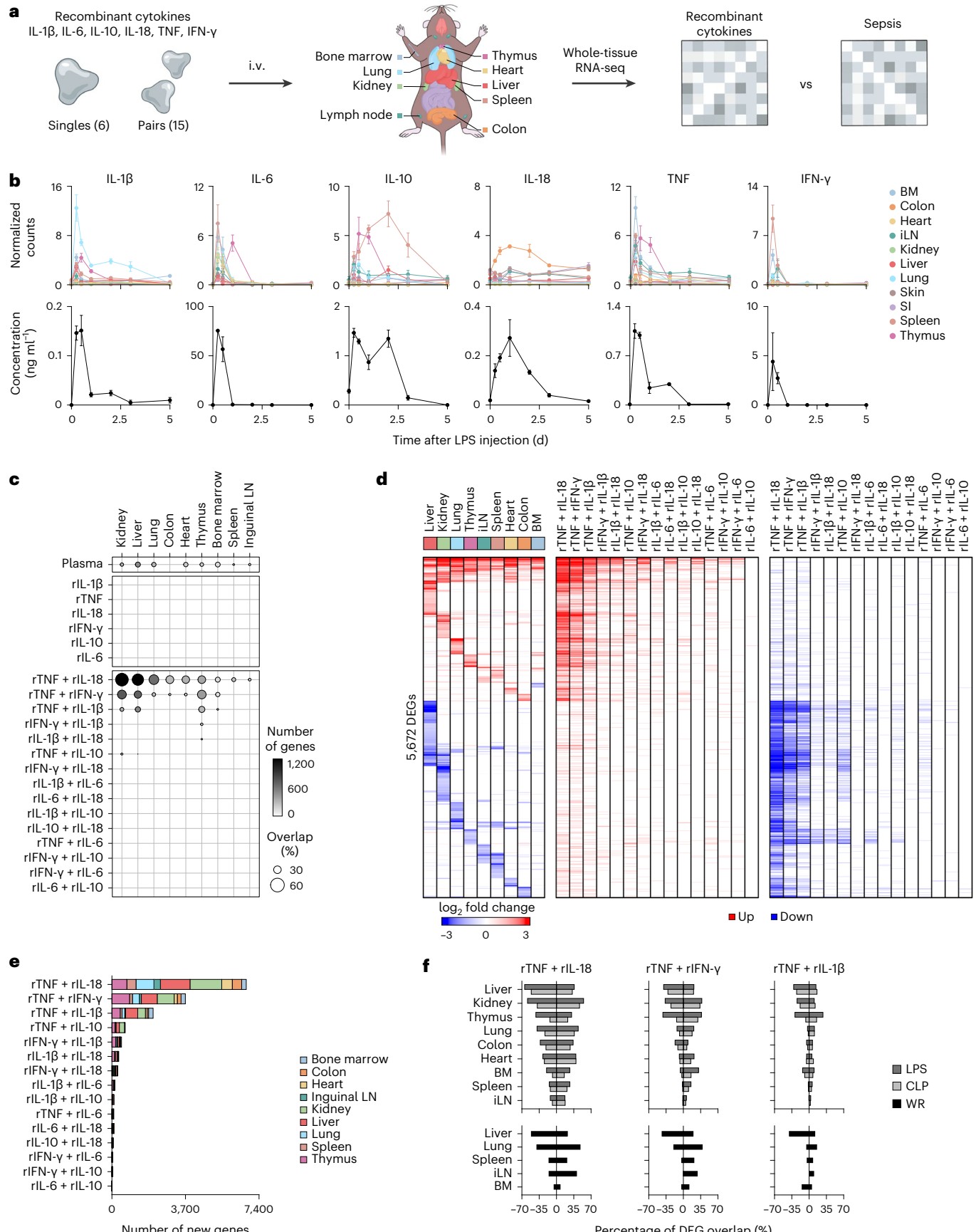

cytokine pairs tested led to synergistic and antagonistic gene expression changes across all nine tissues tested (Extended Data Fig. 4a,b and Supplementary Table 4). For example, liver displayed some of the highest proportions of synergistic and antagonistic genes across all three cytokine pairs: 10.2% and 30.3% for TNF + IL-18, 6.8% and 15.9% for TNF + IFN-γ and 2.7% and 8.7% for TNF + IL-1β, respectively, whereas bone marrow displayed the lowest numbers of synergistic and antagonistic genes (2.1% to 10.3% across all three pairs; Extended Data Fig. 4c). A gene-centric analysis revealed that the pairwise effects of cytokines explained the changes in expression observed for sepsis biomarkers in LPS and CLP sepsis (Extended Data Fig. 4d–g). In addition, these three cytokine pairs regulated genes showing both shared and pair-specific patterns of expression, as in liver and kidney (mostly shared), and heart, spleen and LNs (mostly pair-specific; Extended Data Fig. 5). Moreover, we found that, as in LPS, the tissue effects of TNF plus IL-18, IFN-γ or IL-1β collectively mirrored a large fraction of the effects of CLP and viral sepsis on tissues (Fig. 3f), using previous data on the Western Reserve strain of vaccinia virus[16]. Taken together, these results supported a model whereby nonlinear, pairwise cytokine effects yield tissue states that closely resemble those induced by bacterial and viral sepsis.

## Cytokine pairs explain the physiological effects of sepsis

We investigated the effects of functionally perturbing the four cytokines found to be key to sepsis on tissue states and host physiology and fitness during LPS and CLP sepsis (Fig. 4a). First, we found that TNF deletion or neutralization strongly decreased the number of genes regulated by LPS across tissues, ranging from 9.2% (knockout) and 6.6% (blockade) in thymus to 25.5% (knockout) in liver and 23% (blockade) in kidney (Fig. 4b, Extended Data Fig. 6a,b and Supplementary Table 5a, b). Second, we found that pairwise cytokine perturbations counteracted most of the gene expression changes due to CLP sepsis, with total overlaps in DEGs ranging from 31.9% (2,267/7,106 genes) for anti-TNF + $Il1b^{-/-}$, 45.6% (3,242/7,106 genes) for anti-TNF + $Il18^{-/-}$, to 63.3% (4,497/7,106 genes) for anti-TNF + $Ifng^{-/-}$ (Fig. 4c and Supplementary Table 5c). Interestingly, TNF neutralization alone induced little to no statistically significant changes in tissue expression during CLP, although, in log fold-change space, we observed that many genes showed a trend in expression that was opposite to that of CLP effects without cytokine neutralization (Extended Data Fig. 6c and Supplementary Table 5d). Third, neutralizing antibodies against IL-18, IFN-γ or IL-1β all rescued mice injected with a lethal dose of LPS from a severe body temperature drop, albeit to a lesser extent than blocking TNF alone, which sufficed to completely prevent temperature loss presumably by abrogating key

pairwise cytokine interactions (Fig. 4d). Moreover, blocking TNF or IL-1β led to 100% survival in mice challenged with a lethal dose of LPS, whereas blocking IL-18 or IFN-γ led to partial survival (Fig. 4e). Fourth, $Il18^{-/-}$, $Ifng^{-/-}$ or $Il1b^{-/-}$ mice injected with TNF-neutralizing antibodies kept body temperatures near steady-state levels upon LPS or CLP challenge (Fig. 4f,g). Lastly, we found that injecting recombinant cytokine pairs led to an increase in tissue injury markers in plasma (Fig. 4h) and a drop in body temperature for TNF plus IL-18 or IL-1β (Fig. 4i,j). TNF plus IL-1β displayed a dose-dependent relationship between the quantity of recombinant cytokines administered and the decrease in body temperature and survival of the host (Fig. 4j). Taken together, the similarities in tissue transcriptional states between sepsis and the three key cytokine pairs reflected similarities in physiological effects, including tissue injury, body temperature and survival.

## Cytokine pairs lead to cellular changes mirroring sepsis

The effects of TNF plus IL-18, IFN-γ or IL-1β on tissue states are likely driven by how pairwise cytokine signaling impacts the state and abundance of cell types across organs. For example, all three cytokine pairs led to an increase in the expression level of the neutrophil marker encoded by $S100a8$ in lung, whereas TNF plus IL-18 or IFN-γ decreased the expression of the thymocyte and T cell marker encoded by $Thy1$ in thymus (Extended Data Fig. 4a and Supplementary Table 4). We thus sought to quantify the effects of cytokine pairs and LPS on cell-type abundances across the body (Extended Data Fig. 7a). To infer the abundance of specific cell types from tissue-level measurements (Extended Data Fig. 7b and Supplementary Methods), we computed a cell-type specificity score for each gene expressed in 195 cell types across 9 organs. Resulting gene-centric specificity scores were used to define ranked gene sets for each cell type, which were used to calculate cell-type abundance scores across tissues upon injection of LPS or cytokine pairs (Fig. 5 and Supplementary Table 6). Notably, cytokine pairwise effects on cells mirrored those of LPS in most of the cell types tested and ranged from 23.3% (14/60 cellular effects by LPS at day 0.5) in bone marrow to 100% (42/42 at day 0.5) in kidney, with an average overlap in effects of 48.7% ± 26.9% s.d. across all 9 organs tested (Extended Data Fig. 7c). LPS and cytokine pairs led to several cellular changes, which are well described in sepsis, but lack causal factors. For example, we detected a significant decrease in B and T cell-type scores across lymphoid tissues (spleen, thymus and bone marrow; Fig. 5 and Extended Data Fig. 7d,e), which reflects lymphopenia, a hallmark of sepsis[6]. For T cells, all three cytokine pairs led to a strong decrease in thymocytes as in LPS (Fig. 5 and Extended Data Fig. 7d),

**Fig. 4 | Cytokine pairs explain the physiological and fitness effects of sepsis. a**, Schematic overview of the experimental workflow. The impact of cytokine perturbations using neutralizing antibodies and genetic deletions during LPS or CLP sepsis was assessed by measuring tissue gene expression and host physiological parameters. **b**, Heat maps of DEGs (rows) from whole-tissue mRNA profiles ordered by $k$-means clustering and organ types (top; colors) at 12 h after sublethal LPS injection with or without (control) anti-TNF pretreatment. Values are log₂ fold changes relative to matching organs from untreated mice for LPS without anti-TNF. Statistical analyses were performed with limma (FDR-adjusted $P$ value < 0.01–0.05, absolute fold change > 2). Shown are all the genes found to be differentially regulated in at least one of the indicated conditions (row annotations in black). BM, bone marrow; iLN, inguinal lymph node. **c**, Heat maps of DEGs (rows) from whole-tissue mRNA profiles ordered by $k$-means clustering and organ types (top, colors) at 0.5 d after CLP (severe grade) in wild-type mice injected with isotype control antibodies or $Il18^{-/-}$, $Ifng^{-/-}$ or $Il1b^{-/-}$ mice injected with anti-TNF (left to right). Values are log₂ fold changes relative to matching organs from sham-operated mice for wild-type, or wild-type mice after severe CLP surgeries for $Ifng^{-/-}$, $Il18^{-/-}$ and $Il1b^{-/-}$ mice. Statistical analyses were performed with limma (FDR-adjusted $P$ value < 0.1). Shown are all the genes found to be differentially regulated in at least one of the indicated conditions (row annotations in black). **d**, Measurements of rectal temperature in mice of indicated genotypes with or without indicated neutralizing antibody

pretreatment at 24 h after lethal LPS injection. Statistical differences were determined by one-way analysis of variance (ANOVA) with Tukey–Kramer test. Error bars indicate the s.e.m. ($n = 10$ biologically independent samples for LPS control; $n = 5$ biologically independent samples for other groups). **e**, Survival curves of mice injected with a lethal dose of LPS with or without indicated neutralizing antibody pretreatment ($n = 5$ biologically independent samples). **f,g**, Measurements of rectal temperature in mice of indicated genotypes with or without indicated neutralizing antibody pretreatment at 0.5 d after lethal LPS injection (**f**) or severe CLP surgery (**g**). Statistical differences were determined by one-way ANOVA with Tukey–Kramer test. Error bars indicate the s.e.m. ($n = 4$ biologically independent samples). **h**, Serum levels of indicated organ injury markers at 24 h after injection of a sublethal LPS dose or PBS as control, or 12 h after injection of indicated recombinant cytokine pairs. Statistical differences were determined by one-way ANOVA with Tukey–Kramer test. Error bars indicate the s.e.m. ($n = 4$ biologically independent samples). ALT, alanine transaminase; BUN, blood urea nitrogen. **i**, Measurement of rectal temperature at 16 h after injection of a sublethal LPS dose, indicated cytokines or PBS as control. Error bars indicate the s.e.m. ($n = 4$ biologically independent samples). **j**, Measurements of rectal temperature ($y$ axis; left and right) relative to time after injection (left) or varying doses (right, $x$ axis) of recombinant (r)IL-1β in combination with rTNF (1 μg). Error bars indicate the s.d. ($n = 2$ biologically independent samples).

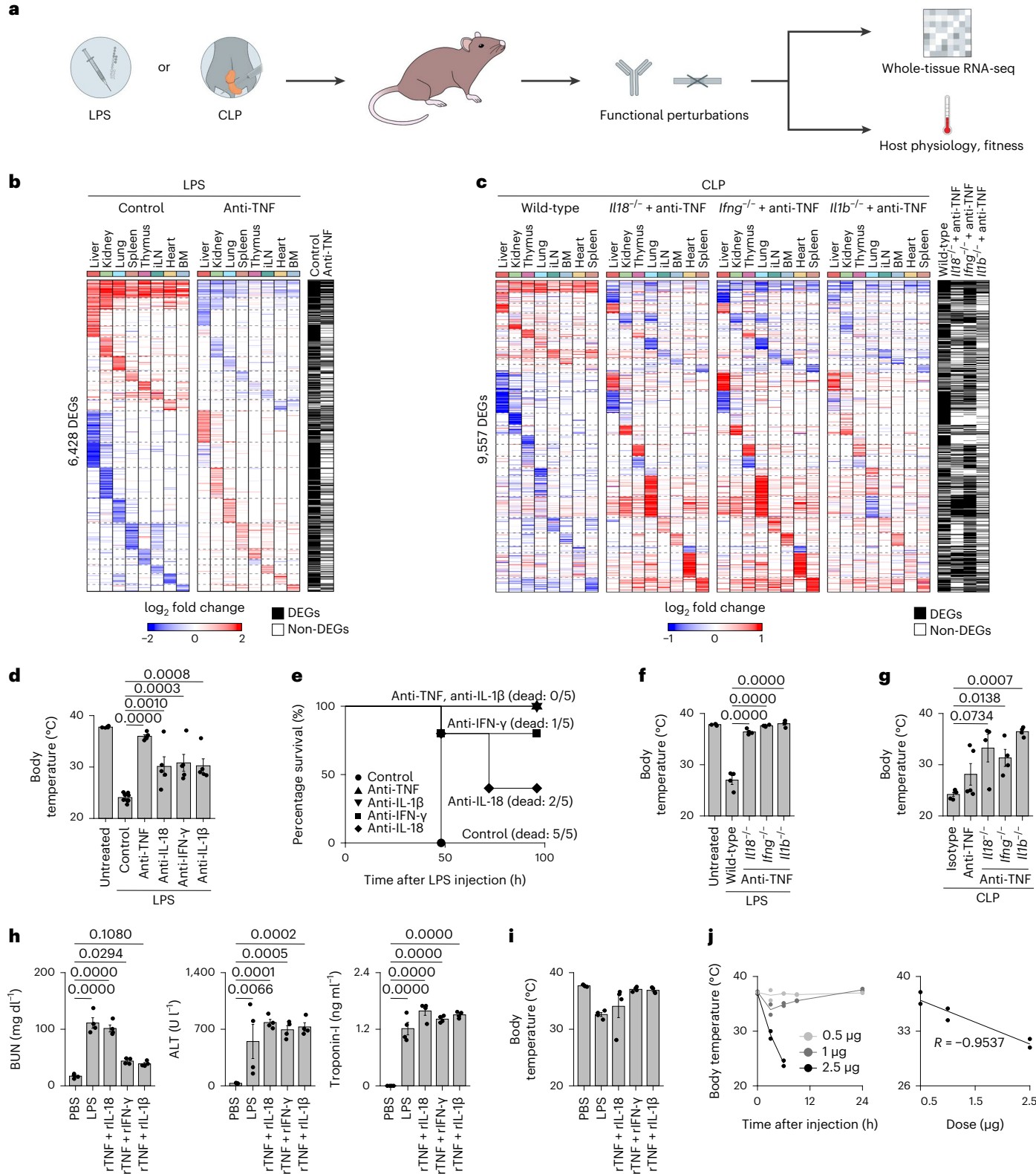

which is in agreement with the well-described phenomena of T cell depletion and thymic involution during sepsis[9]. However, cytokine pairs did not mirror the effects of LPS on splenic T cells (Fig. 5 and Extended Data Fig. 7d). For B cells, TNF plus IL-18 led to a decrease in several splenic B cell types, whereas in the bone marrow, none of the three pairs tested recapitulated LPS effects on B cells (Fig. 5 and Extended Data Fig. 7e). Lastly, LPS and cytokine pairs led to an increase

in abundance of endothelial cell types associated with the heart, kidney and liver (Fig. 5 and Extended Data Fig. 7f), which is corroborated by recent work[27], and our results identify the cytokine factors driving this effect on the endothelium across tissues. Overall, these results provide an organism-wide view of the impact of cytokine pairs and LPS at the cellular level and a mechanistic basis for both well-described and less-studied cellular phenomena in sepsis.

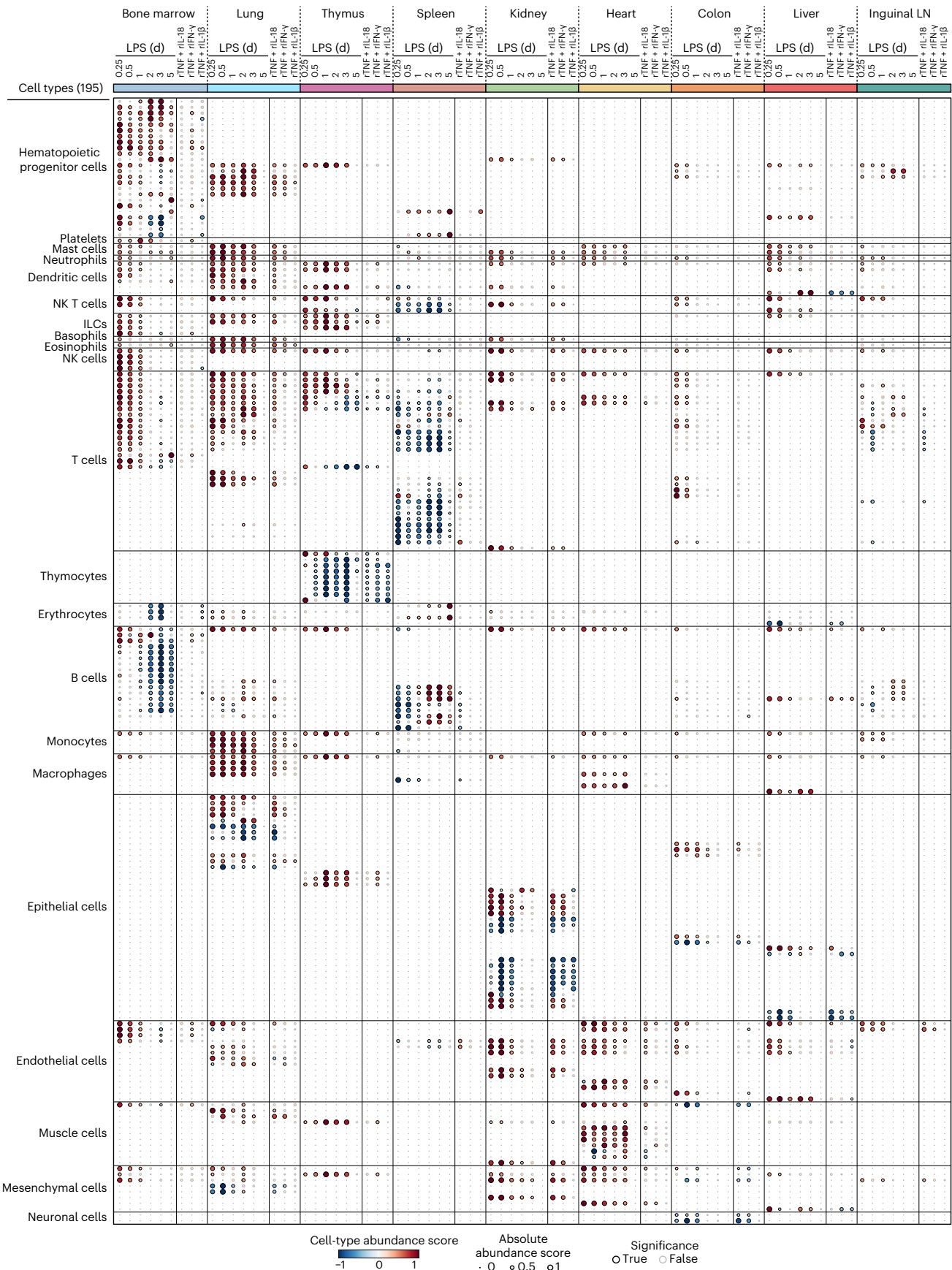

**Fig. 5 | The pairwise effects of TNF plus IL-18, IFN-γ or IL-1β mirror the cellular effects of sepsis across organs.** Whole-tissue RNA-seq profiles were integrated with cell-type-specific gene sets (Supplementary Methods) to obtain cell-type abundance scores computed for 195 cell types (rows) across 9 tissues (colors; top) upon injection of a sublethal dose of LPS or indicated recombinant cytokine pairs (columns). Cell-type abundance score, absolute abundance score and significance (absolute value of z-score > 1) are shown as colors, circle size and outline of the circle, respectively. Individual scores can be found in Supplementary Table 6. LN, lymph node; ILC, innate lymphoid cell; NK, natural killer.

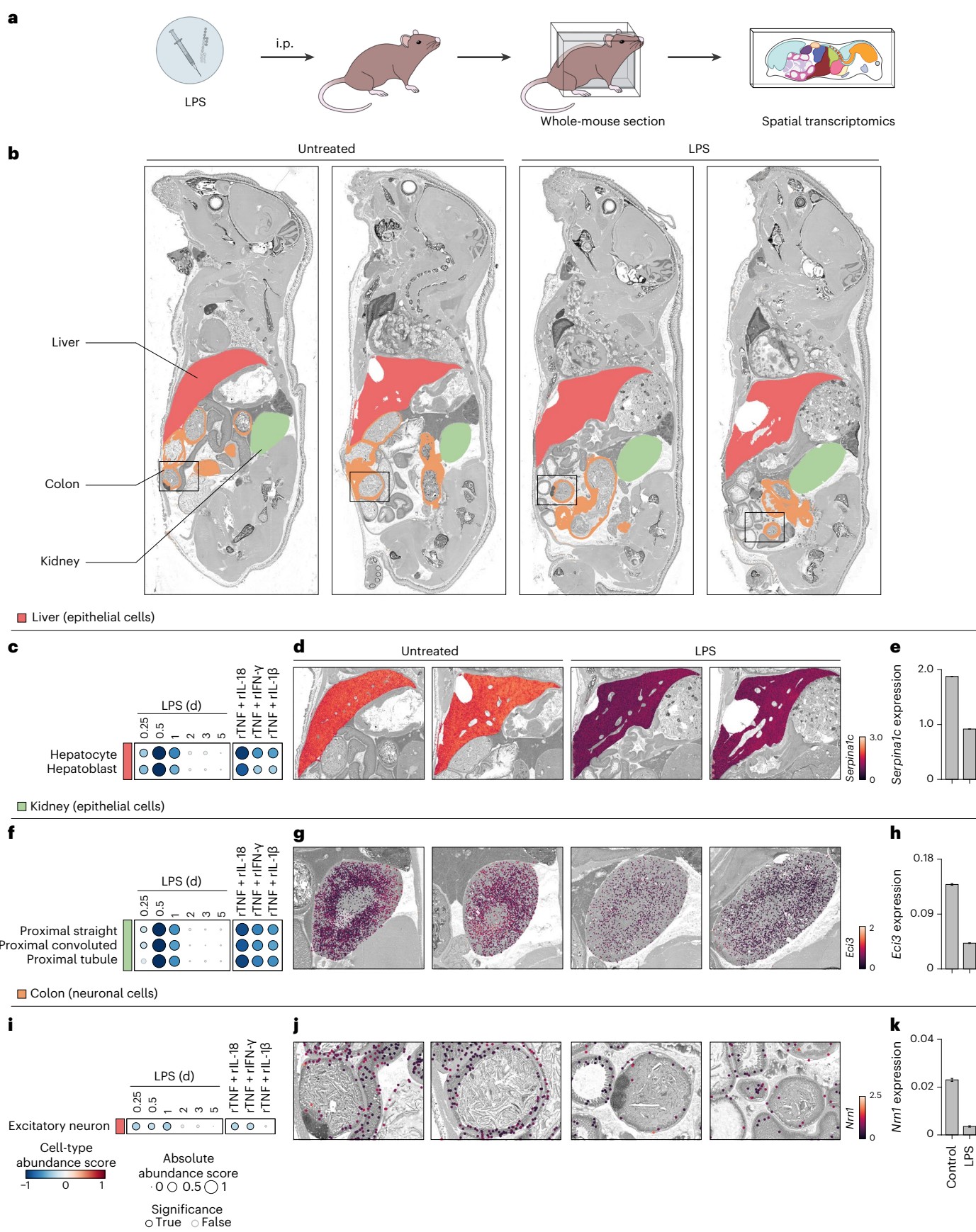

**Fig. 6 | TNF plus IL-18, IFN-γ or IL-1β are responsible for the cellular effects of sepsis on epithelial and neuronal cells across tissues. a,** Schematic overview of the experimental workflow for whole-mouse sectioning followed by large-format spatial transcriptomics. **b,** Whole-mouse spatial transcriptomics analysis of indicated tissue clusters overlaid on a grayscale hematoxylin and eosin (H&E) staining. Shown are whole-mount sections and spatial transcriptomics data from 5-week-old mice injected with a sublethal dose of LPS (5 mg per kg body weight) or left untreated as control. **c,f,i,** Cell-type abundance scores computed for indicated cell types (rows) and tissues (colors) upon injection of a sublethal dose of LPS

in wild-type (left) or injected with indicated recombinant cytokine pairs (right; columns). Black borders indicate significance (*z*-score > 1). **d,e,g,h,j,k,** Whole-mouse spatial transcriptomics data (**d, g** and **j**) from control and LPS conditions (columns) were magnified to show liver (**d**), kidney (**g**) and colon (**j**) tissues. Normalized expression for cell-type marker genes *Serpina1c*, *Eci3* or *Nrn1* was overlaid as on a grayscale H&E image. Bar plots (**e, h** and **k**) of average expression of indicated genes across all spatial transcriptomics array spots covering indicated tissues. Error bars indicate the s.e.m. (*n* > 10, the number of spatial transcriptomics array spots covering indicated tissues). i.p., intraperitoneal.

## Spatial analysis validates the cellular effects of cytokine

We aimed to validate experimentally the functional associations between LPS and cytokine pairs and changes in cell-type abundances that were predicted by our computational analyses. We measured changes in gene expression across whole-mount sections using a custom, large-format spatial transcriptomics method (Fig. 6a). First, we focused on epithelial and neuronal cell types in liver, kidney and colon tissues and found that marker genes for hepatocytes, kidney epithelia and colon neurons were downregulated by LPS sepsis, in agreement with our computational scoring method (Fig. 6b–k and Extended Data Fig. 8a,b). We also showed that hepatocytes were negatively impacted by LPS and cytokine pairs using TUNEL staining (Extended Data Fig. 8c,d). In kidney, we confirmed the prediction that LPS negatively impacted proximal tubule epithelial cells using a commercial spatial transcriptomics platform (Extended Data Fig. 8e–k). Second, we monitored changes in immune cells across the body. We found neutrophil accumulation in all 17 tissues profiled by whole-mouse sections and in the vasculature, as indicated by the upregulation of *S100a8* transcripts, a neutrophil marker gene (Fig. 7a–c). These results were corroborated in lung using immunohistochemistry, where we observed a higher recruitment of neutrophils upon recombinant TNF plus IL-18 or IFN-γ injection than in LPS (Fig. 7d,e). Macrophages were also found to be upregulated across tissues (Extended Data Fig. 9a), in agreement with previous work on a subset of tissues[23], which we validated by whole-mouse profiling of *Marco* and immunohistochemistry for the macrophage marker F4/80 (Extended Data Fig. 9b–e). In spleen, we found that TNF combined with IL-18 to deplete B cell subsets including follicular and, even more so, marginal zone B cells, by spatial transcriptomics (Fig. 7f–h) and flow cytometric analysis (Fig. 7i,j). These effects on splenic B cells were in agreement with work using CLP[28], although the causal factors for this phenotype were not previously known. In the bone marrow, we confirmed that TNF plus IL-1β are sufficient to decrease the abundance of cell types from the erythroid lineage (Extended Data Fig. 9f–h), which help to explain anemia, a well-described phenomenon in sepsis. Lastly, we found that injecting LPS in *Il18*⁻/⁻, *Ifng*⁻/⁻ and *Il1b*⁻/⁻ mice treated with TNF-neutralizing antibodies abrogated the cellular effects validated above in all cases but lung granulocytes, suggesting that other pathways are likely at play for specific processes triggered by sepsis (Extended Data Fig. 10).

Overall, by mapping the effects of LPS and cytokine pairs on cell-type abundances, we provided a mechanistic basis for known

and previously unreported cellular effects of sepsis on tissues. For example, the relative abundance scores of immune cell types are positively and negatively regulated by at least one of the three cytokine pairs across all nine organs tested (Fig. 8). While endothelial cell types were mostly upregulated, epithelial and mesenchymal cell types were equally upregulated or downregulated across tissues (Fig. 8). Of the three recombinant cytokine pairs tested, TNF plus IL-18 was the one impacting the most cell types across the most tissues, reflecting its wider impact on tissue transcriptional states compared to the other two cytokine pairs at the doses tested (Fig. 8). Taken together, our data uncover a pairwise cytokine code that explains most of the host response to sepsis ranging from genes to cells to tissue physiology and host fitness.

## Discussion

While sepsis remains a leading cause of death in intensive care units worldwide, our understanding of the pathogenesis of sepsis across most tissues and organs of the body is lacking. To begin to address this fundamental gap in knowledge, we mapped the organismal response to sepsis over time by measuring changes in gene expression across tissues in mouse models of the disease. Using cytokine injections and perturbations in vivo, we discovered a hierarchical cytokine module composed of TNF, IL-18, IFN-γ and IL-1β that sufficed to explain most of the organism-wide response to sepsis, ranging from genes to cells to tissue physiology and host fitness. Our work decodes the chaos in systemic cytokine signaling during sepsis into a simplifying, pairwise cytokine message and provides spatiotemporal data key to build a mechanistic framework for the impact of sepsis on the whole organism.

What did organism-wide maps of gene expression tell us about sepsis? First, our data revealed a plethora of changes that come with the initiation and resolution of sepsis, both at the molecular and cellular levels. These changes were detected in all organ systems tested and encompassed most known, if not all, biomarkers and physiological events linked to sepsis. For example, of the 872 genes with a PubMed Gene Reference into Function (geneRIF) annotation containing the keyword 'sepsis', 69.7% (608/872 genes as of 18 March 2023) were regulated in at least one tissue and time point in our LPS and CLP sepsis data. Future work is needed to elucidate which regulated genes are causal or bystander and beneficial or detrimental during sepsis.

**Fig. 7 | The impact of TNF plus IL-18, IFN-γ or IL-1β on hematopoietic cell types provides a mechanistic basis for sepsis effects on the immune system. a,f,** Cell-type abundance scores computed for indicated cell types (rows) and tissues (colors) upon injection of a sublethal dose of LPS in wild-type (left) or injected with indicated recombinant cytokine pairs (right; columns). Black borders indicate significance (*z*-score > 1). **b,c,** Whole-mouse spatial transcriptomics analysis (**b**) of *S100a8* mRNA levels overlaid on a grayscale H&E staining. Shown are whole-mount sections and spatial transcriptomics data from 5-week-old mice injected with a sublethal dose of LPS (5 mg per kg body weight) or left untreated as control. Bar plot (**c**) of average expression of *S100a8* across all spatial transcriptomics array spots covering indicated tissues from Fig. 7b. Error bars indicate the s.e.m. (*n* > 10, the number of spatial transcriptomics array spots covering indicated tissues). **d,e,** Images

(×40 magnification; **d**) from Ly6G immunohistochemistry in lungs from mice injected with LPS, indicated cytokines or left untreated as controls. Bar graph (**e**) shows quantifications of Ly6G⁺ cells per field of view. Scale bars, 100 μm. Error bars indicate the s.e.m. (*n* = 10 independent field of view). **g,h,** Whole-mouse spatial transcriptomics data (**g**) from control and LPS conditions (columns) were magnified to only show spleen tissue. *Cr2* normalized expression was overlaid as cell-type markers on a grayscale H&E image. Bar plot (**h**) of average expression of *Cr2* across all spatial transcriptomics array spots covering indicated tissues. Error bars indicate the s.e.m. (*n* > 10, the number of spatial transcriptomics array spots covering indicated tissues). **i,j,** Flow cytometry analysis (**i**) of splenic B cells from mice injected with a sublethal dose of LPS or indicated cytokines. Bar graphs (**j**) show quantifications in absolute count per tissue. Error bars indicate the s.e.m. (*n* = 2 biologically independent samples).

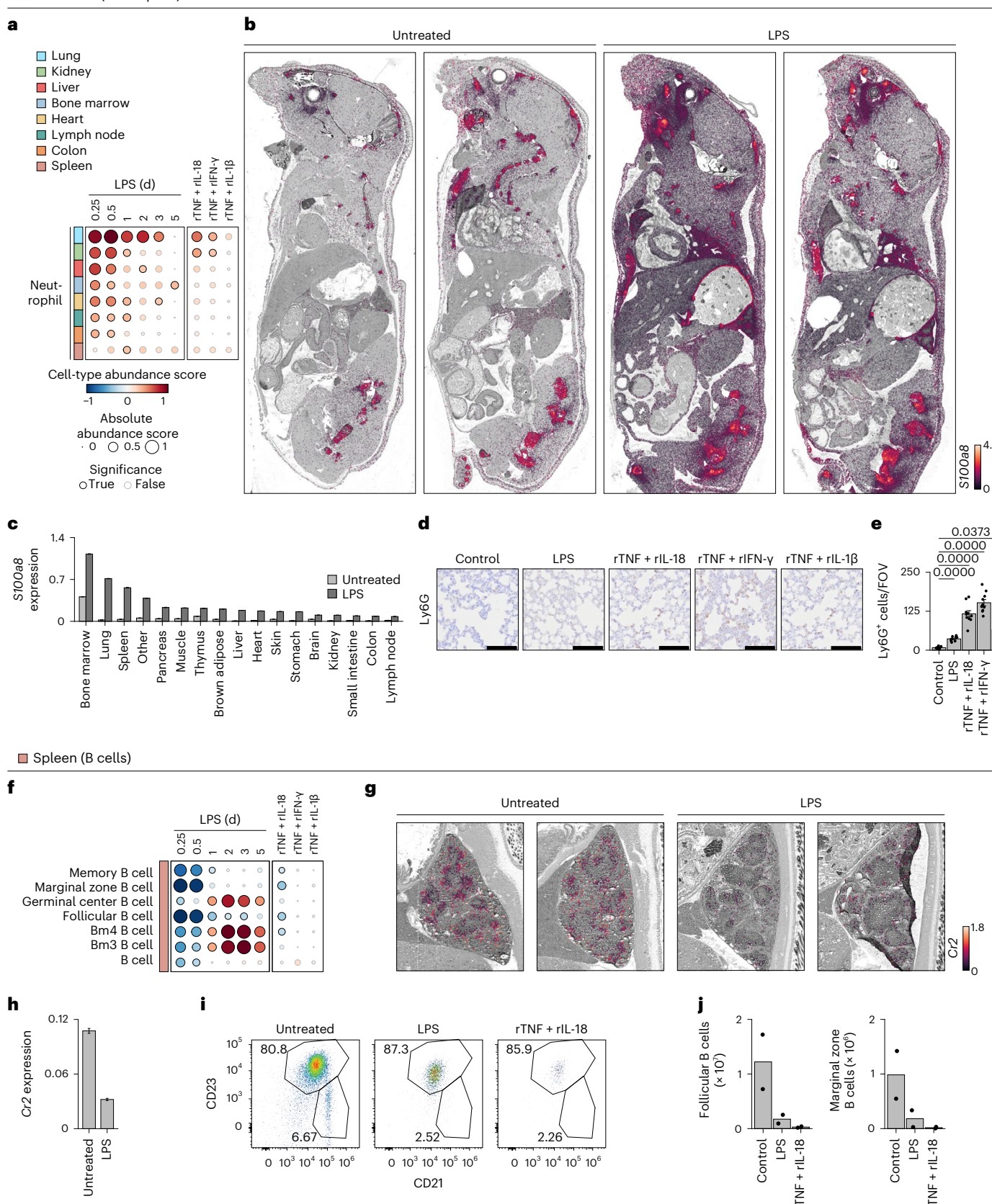

□ All tissues (neutrophils)

□ Spleen (B cells)

Second, we found that nonlymphoid tissues regained homeostasis sooner than lymphoid ones in endotoxemia. This result is reminiscent of how some organs reverse dysfunction in sepsis, including those poor at regenerating such as heart, lung, kidney or brain, whereas the immune system suffers long-term dysregulation with life-threatening consequences for survivors[14,18]. Further mining of our data might help to identify factors that safeguard nonlymphoid tissues, such as IL-10 for microglia[29] or GDF15 for heart[30] which are both present in our data, or

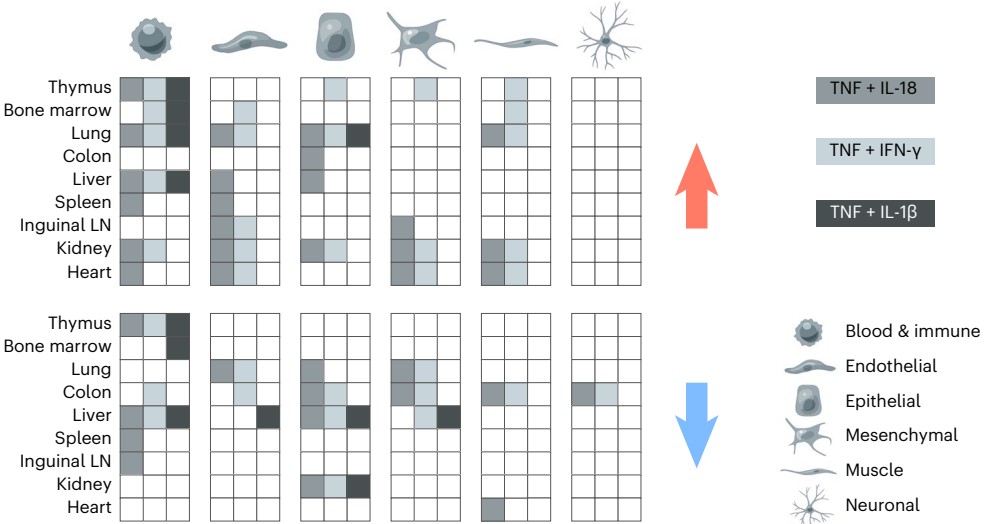

**Fig. 8 | A pairwise cytokine code explains the organism-wide effects of sepsis on core cell types.** Schematic, qualitative summary of the impact of the three cytokine pairs indicated (gray scale) on the indicated core cell types (columns and bottom-right legend) across tissues (rows). Red and blue arrows indicate an increase and decrease, respectively, in cell-type abundance score for each cytokine pair on each core cell type in each tissue. LN, lymph node.

those that damage lymphoid tissues and cells, such as pairwise effects in the cytokine network. Future work is needed to assess the dynamics of tissue recovery, if any, during CLP sepsis with or without antibiotics treatment mimicking human patient treatment regimens.

Third, we built an organism-wide map of sepsis effects at the resolution of cell types by computing abundance scores for 195 cell types across 9 organ types. In addition to revealing the scope of the cellular effects of sepsis on tissues, our analysis provided a causal linkage between specific cytokine pairs and cell types across tissue contexts. Notably, all seven associations between cytokines and cell types selected for further study were validated by experiments, encompassing hepatocytes, kidney epithelia, colon neuronal cells, splenic B cells, bone marrow erythroid cells and whole-body neutrophils and macrophages. Most of these cellular effects were previously observed in sepsis or endotoxemia, such as an increase in thymic macrophages[31], erythropenia[20], splenic B cell loss[9] or changes in kidney tubules[32] but, crucially, lacked causal factors. Future work is needed to test other predictions and define the mechanisms underlying cellular changes, such as alterations in the proliferation, death, migration or intracellular state of the cells affected by sepsis. Taken together, our spatiotemporal data provide detailed insights in the quest toward defining a mechanistic framework to explain sepsis.

What sense can be made of the cytokine cacophony taking place in the blood during sepsis? While uncontrolled cytokine signaling is harmful to the body, we lack knowledge about which cytokine signaling events impact which cell types in the context of which tissues. By measuring the impact of six cytokines alone or in pairwise combinations on tissue mRNA expression profiles, we captured the net output of tissue-level responses to cytokine inputs—as opposed to focusing on the response of a single cell type. Our data support a model whereby a few elements of cytokine information—TNF plus IL-18, IFN-γ or IL-1β— suffice to explain a large fraction of the molecular and cellular effects of sepsis across tissues. In addition, this model reveals the existence of a simplifying hierarchy among the cytokines upregulated in blood during sepsis, which will help to build a unifying mechanistic framework for sepsis and other cytokine storms[7]. Notably, the proposed cytokine hierarchy relies on nonlinear interactions between TNF and IL-18, IFN-γ or IL-1β signaling, a notion well supported by four decades of work on cytokine interactions in vitro and in vivo. For example, TNF has been shown to combine synergistically or antagonistically with IFN-γ or IL-1β to impact secretion, cell death or proliferation and cell states in immune

and nonimmune cells in culture[33–37]. While the interaction between TNF and IL-18 had not been reported to our knowledge, TNF plus IFN-γ[38–41] or IL-1β[42–44] worsen the outcome of sepsis and other inflammatory disorders in vivo. The cytokines of this module also influence each other's production[7], which further supports the hierarchy uncovered by our pairwise cytokine screening data.

Future investigations are needed to define the direct and indirect effects of each cytokine pair on each cell type. For example, it is likely that some cytokine pairwise effects act through downstream factors, including through the release of other cytokine or non-cytokine diffusible factors that are directly sensed by the cells and tissues. In addition, while our data linked one of the three cytokine pairs or more with 52% (178/342 at day 0.5) of the target cell types tested in at least one organ type and impacted with LPS, the other half of the cellular effects of LPS on tissues remained unexplained by the three cytokine pairs used here. Thus, further work on other cytokines and non-cytokine factors, such as the complement or coagulation systems, is needed to pinpoint the causative factors responsible for the observed cellular effects of sepsis on tissues. The detailed signaling events mediating cytokine interactions at the level of cells also remain to be elucidated, such as the putative rewiring of the MAPK, NF-kB, IRF and Jak/STAT pathways that have previously been linked to the interaction between TNF and IFN-γ[36,39,45,46].

Why is TNF the central node of this cytokine module recapitulating many of the effects of sepsis? After half a century since the first isolation of TNF as a factor that could kill tumor cells[47], TNF has been implicated in the pathogenesis of countless infectious and noninfectious diseases[48,49]. In sepsis, TNF is one of the earliest cytokines produced in mice and humans, peaking in the blood in less than 2 h with a circulating half-life of less than 20 min in mice[50,51]. TNF antibodies protect against lethal sepsis when present before or early on upon the start of the disease[52,53], but not later in the disease, which helps to explain the failure of anti-TNF therapy in humans with sepsis[2,54]. Interestingly, pretreatment with anti-TNF leads to beneficial effects in humans, such as in the suppression of the Jarisch–Herxheimer reaction occurring in response to antibiotic treatment of louse-borne relapsing fever[55,56]. Conversely, the infusion of recombinant TNF in humans suffices to trigger flu-like symptoms[57]. Lastly, our findings about the central role of TNF in a cytokine circuit controlling sepsis are reminiscent of the existence of a cytokine hierarchy defining human chronic inflammatory diseases across tissues[58]. Inhibiting TNF has shown remarkable therapeutic benefits

in patients with psoriasis, psoriatic arthritis, Crohn's disease, ulcerative colitis, ankylosing spondylitis, juvenile arthritis and many other less prevalent diseases. However, targeting cytokines such as IL-6, IL-1 or IL-17/IL-23 showed a much narrower range of efficacy, suggesting that TNF combines with select cytokines in select organs in the pathogenesis of inflammatory disorders[58]. The multi-tissue effects of TNF in inflammation are likely a product of the existence of numerous TNF receptors that are ubiquitously expressed[49]. Taken together, these observations in sepsis and beyond help to contextualize the organism-wide effects of the three TNF-centered cytokine pairs identified by our data as critical to explain sepsis.

Overall, our work provides fundamental insights to help build a mechanistic framework explaining the organism-wide effects of sepsis, which will fuel therapeutic innovation for a disease lacking targeted drugs.

## Online content

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

## Methods

### Mice

Female C57BL/6J mice (wild-type, stock 000664), B6.129S7-*Ifng*[tm1Ts]/J (Ifng KO, stock 002287), B6.129P2-*Il18*[tm1Aki]/J (Il18 KO, stock 004130), C57BL/6J-*Il1b*[em2Lutzy]/Mmjax (Il1b KO, stock 068082-JAX) and B6.129S-*Tnf*[tm1Gkl]/J (Tnf KO, stock 005540) were obtained from the Jackson Laboratories. Animals were housed in specific pathogen-free and BSL2 conditions at The University of Chicago, and all experiments were performed in accordance with the US National Institutes of Health Guide for the Care and Use of Laboratory Animals and approved by The University of Chicago Institutional Animal Care and Use Committee.

### Mouse models of endotoxemia and sepsis

For LPS endotoxemia, mice were injected intraperitoneally with either lethal (10–15 mg per kg body weight) or sublethal (3–5 mg per kg body weight) doses of LPS derived from *Escherichia coli* O55:B5 (Sigma-Aldrich) diluted in PBS. Dosing was established for each lot of LPS by in vivo titration. CLP was performed as described by others[59,60]. Briefly, mice were anesthetized with isoflurane. A 1- to 2-cm midline laparotomy was performed and the cecum was exposed. The cecum was ligated with 6-0 silk sutures (Ethicon) and perforated as follows to vary disease severity: (1) mild sepsis: ligate at distal 33% position and perforate once with a 21-gauge needle; (2) moderate sepsis: ligate at distal 40% position and perforate twice with a 19-gauge needle; (3) severe sepsis: ligate immediately below the ileocecal valve and perforate twice with a 19-gauge needle. The cecum was tucked back into the peritoneum and gently squeezed to extrude a small amount of fecal content. The peritoneal wall was closed using absorbable suture. The skin was closed with surgical staples. To resuscitate animals, 1 ml of saline was injected subcutaneously. Mice were temporarily placed on a heating pad for recovery. Sham-operated mice underwent the same procedure except that the cecum was neither tied nor perforated.

### Recombinant cytokine injections

C57BL/6J mice were injected intravenously with 2.5 µg of recombinant TNF, IL-1β, IL-6, IL-10, IL-18 or IFN-γ used alone (6 singles) or in pairwise combinations (15 pairs).

### Neutralizing antibody and drug treatments

For neutralizing antibodies, C57BL/6J or indicated knockout mice were injected intraperitoneally with 50 µg of TNF (clone BE0058, BioXCell), IL-18 (clone BE0237, BioXCell), IFN-γ (clone BE0055, BioXCell) or IL-1β (clone BE0246, BioXCell) neutralizing antibodies in 100 µl of PBS 1 h before LPS injection.

### Blood analysis

Mouse whole blood was harvested by cardiac puncture and plasma and serum were isolated using lithium heparin-coated Microtainer blood collection tubes (BD, 365965) and Microtainer blood collection tubes (BD, 365978), respectively. For flow cytometric, bead-based immunoassays, plasma was diluted and processed using the LEGENDplex Mouse Inflammation Panel (BioLegend, 740446) and Mouse Macrophage/Microglia Panel (BioLegend, 740846) kits. Data were acquired on the NovoCyte flow cytometer (Acea Biosciences/Agilent) and analyzed using the LEGNEDplex software v8 (BioLegend). To measure tissue injury marker levels in sera, samples were processed with the following kits for BUN (BioAssay Systems DIUR-100), ALT (Cayman Chemical, 700260) and troponin-I (Life Diagnostics, CTNI-1-HS) levels according to the manufacturer's instructions.

### Tissue harvest

Tissues were harvested, frozen and stored as previously described[16,17]. Mice were anesthetized with 2,2,2-tribromoethanol (250–500 mg per kg body weight) and perfused transcardially with PBS containing 10 mM EDTA (to avoid signal contamination from blood in tissues).

Before perfusion, blood was collected by cardiac puncture and stored on ice and, immediately after perfusion, tissues were placed in RNA-preserving solution (5.3 M ammonium sulfate, 25 mM sodium citrate, 20 mM EDTA) and kept at 4 °C overnight before transfer at −80 °C for storage. For each mouse, we harvested up to 13 tissues in total: iLNs, flank skin, thymus, heart, lung, spleen, kidney, small intestine, colon, liver, brain, bone marrow and PBMCs. Small intestine and colon were cut longitudinally and washed extensively in PBS to completely remove feces contamination. Bone marrow cells were collected from femora and tibiae, stored overnight in RNA-preserving solution at 4 °C, centrifuged at 5,000*g* for 5 min at 4 °C and cell pellets were stored at −80 °C.

### Whole-tissue RNA extraction

Whole-tissue RNA extraction was performed as described previously[17]. Briefly, tissues stored in RNA-preserving solution were thawed and transferred to 2-ml tubes containing 700–1,500 µl (depending on tissue) of PureZOL (Bio-Rad, 7326890) or homemade TRIzol-like solution (38% phenol, 0.8 M guanidine thiocyanate, 0.4 M ammonium thiocyanate, 0.1 M sodium acetate, 5% glycerol). Tissues were lysed by adding 2.8-mm ceramic beads (OMNI International, 19–646) and running 1–3 cycles of 5–45 s at 3,500 r.p.m. on the PowerLyzer 24 (QIAGEN). For liver, brain and small intestine samples, tissues were lysed with 3–5 ml using M tubes (Miltenyi Biotec, 130-096-335) and running 1–4 cycles of the RNA_02.01 program on the gentleMACS Octo Dissociator (Miltenyi Biotec). Next, lysates were processed in deep 96-well plates (USA Scientific, 1896–2000) by adding chloroform for phase separation by centrifugation, followed by precipitation of total RNA in the aqueous phase using magnetic beads coated with silane (Dynabeads MyOne Silane; Thermo Fisher Scientific, 37002D), buffer RLT (QIAGEN, 79216) and ethanol. Genomic DNA contamination was removed by on-bead DNase I (Thermo Fisher Scientific, AM2239) treatment at 37 °C for 20 min. After washing steps with 80% ethanol, RNA was eluted from beads. This RNA extraction protocol was performed on the Bravo Automated Liquid Handling Platform (Agilent)[17]. Sample concentrations were measured using a Nanodrop One (Thermo Scientific). RNA quality was confirmed using a Tapestation 4200 (Agilent Technologies). The samples with low RNA quality were excluded from the subsequent experiments.

### RNA sequencing

For each tissue sample, full-length cDNA was synthesized in 20 µl final reaction volume containing the following: (1) 10 µl of 10 ng µl⁻¹ RNA; (2) 1 µl containing 2 pmol of a custom RT primer biotinylated in 5′ and containing sequences from 5′ to 3′ for the Illumina read 1 primer, a 6-bp sample barcode (up to 384), a 10-bp unique molecular identifier (UMI) and an anchored oligo(dT)₃₀ for priming[61]; and (3) 9 µl of RT mix containing 4 µl of 5× RT buffer, 1 µl of 10 mM dNTPs, 2 pmol of template switching oligo and 0.25 µl of Maxima H Minus Reverse Transcriptase (Thermo Scientific, EP0753). First, barcoded RT primers were added to RNA, which were then denatured at 72 °C for 1 min followed by snap cooling on ice. Second, the RT mix was added and plates were incubated at 42 °C for 120 min. For each library, double-stranded cDNA from up to 384 samples were pooled using DNA Clean & Concentrator-5 columns (Zymo Research, D4013) and residual RT primers were removed using exonuclease I (New England Biolabs, M0293). Full-length cDNAs were amplified with 5 to 8 cycles of single-primer PCR using the Advantage 2 PCR Kit (Clontech, 639206) and cleaned up using SPRIselect magnetic beads (Beckman Coulter, B23318). cDNA was quantified with a Qubit dsDNA High Sensitivity Assay Kit (Thermo Fisher Scientific, 32851) and 50 ng of cDNA per pool of samples was tagmented using the Tagment DNA Enzyme I (Illumina, 20034197) and amplified using the NEBNext Ultra II Q5 Master Mix (New England BioLabs, M0544L). Libraries were gel purified using 2% E-Gel EX Agarose Gels (Thermo Fisher Scientific, G402002), quantified with a Qubit dsDNA High Sensitivity Assay Kit (Thermo Fisher

Scientific, Q32851) and a Tapestation 4200 (Agilent Technologies) and sequenced on the NextSeq 550 platform (Illumina).

### Custom, whole-mouse spatial transcriptomics using Array-seq

Mice injected with LPS or left untreated as control were euthanized with $CO_2$, frozen in a dry ice–hexane bath after removing all body hair and teeth and stored at −80 °C until use. Frozen section preparation and section transfer were carried out by modifying Kawamoto's film method[62–64]. Frozen mice were embedded in a cryo-embedding medium and sectioned (10-μm thickness) using a Leica CM3600-XP cryomacrotome. Resulting whole-mouse sections were transferred onto custom, large-format microarrays (30-μm spot diameter with 36.65 μm center-to-center distance between spots), which were repurposed for spatial transcriptomics measurements using the Array-seq method. After transfer, sections were fixed in methanol, stained with H&E and imaged on an Olympus VS2000 slide scanner (×20 magnification). Sections were permeabilized (1% pepsin), incubated for in-tissue reverse transcription and treated with proteinase K for tissue removal. Resulting full-length, single-stranded cDNAs were denatured and retrieved from the array using potassium hydroxide and purified by column clean up (Zymo Research). cDNA was processed for single-primer PCR amplication followed by sequencing library construction using tagmentation (Nextera DNA Library Prep Kit) and final PCR amplification. Resulting libraries were sequenced on the NovaSeq 6000 (Illumina) and sequencing data was preprocessed using STAR/STARsolo (version 2.7.10a)[79] (https://github.com/alexdobin/STAR/blob/master/docs/STARsolo.md/) for read alignment using the GRCm39 mouse reference genome, spatial barcode demultiplexing and UMI counting. Resulting spatial transcriptomics data was normalized, processed for differential expression analysis, and visualized using custom Python 3.8.5 (https://www.python.org/) scripts and existing packages, including Scanpy (version 1.9.1)[65], scikit-image (version 1.1.3)[66] and Seaborn (version 0.11.2)[67,68] and scikit-learn (version 0.24.2). Cell-type deconvolution for each spatial transcriptomics spot was done using the CARD package (version 1.0.0)[69].

### Commercial, kidney spatial transcriptomics using Visium

Mouse kidneys were dissected from LPS-injected or control mice without transcardial perfusion and frozen in optimal cutting temperature media. In total, 10-μm frozen tissue sections were cut with a CM1850 Cryostat (Leica) and mounted onto a Visium Spatial Gene Expression library preparation slide (10x Genomics). Samples were processed to generate spatial transcriptomics sequencing libraries according to the manufacturer's instructions. In brief, sections were fixed in 100% methanol and stained with H&E reagents. H&E-stained sections were imaged using a CRi Panoramic MIDI Whole Slide Scanner with ×20 magnification. Sections were then permeabilized with 0.1% pepsin in 0.01 M HCl for 14 min at 37 °C and processed for in-tissue reverse transcription followed by on-slide second-strand synthesis. Resulting cDNA was used to construct sequencing libraries that were sequenced on the NextSeq 550 platform (Illumina), with 28 bases for read 1 and 56 for read 2 and at a depth of 78–114 million total reads per sample. The output data of each sequencing run (Illumina BCL files) were converted into FASTQ files using Bcl2Fastq v.2.19.1. The Space Ranger software (v.1.2.0, 10x Genomics) was used to process, align and summarize the FASTQ files against a GRCm39 mouse reference genome. Raw UMI count spot matrices, spot coordinates and images were imported into Python using Scanpy (v.1.9.1)[65]. Raw UMI counts were log10 normalized and clustered using a Louvain algorithm (resolution of 0.35). Differential expression between control and LPS-treated samples was performed using Scanpy's rank_genes_groups function using a Wilcoxon rank-sum test. Spatially resolved counts of differentially expressed genes were overlaid with corresponding grayscale H&E images and visualized using Seaborn v.0.11.2 (https://github.com/mwaskom/seaborn/).

### Histology

Tissue processing, embedding, sectioning, immunohistochemistry using purified mouse Ly6G (clone 1A8, BioLegend) and F4/80 (clone BM8, BioLegend) antibodies, or TUNEL (terminal deoxynucleotidyl transferase dUTP nick end labeling) staining was performed by the Human Tissue Resource Center at the University of Chicago. Section images were obtained using the Slideview VS200 Research Slide Scanner (Olympus). Image analysis and quantification (Ly6G$^+$, TUNEL$^+$ and F4/80$^+$ areas) were performed using ImageJ (https://imagej.nih.gov/ij/).

### Flow cytometry

To analyze splenic B cells, total splenocytes were obtained by mashing spleens on 70-μm filters followed by red blood cell lysis (Lonza). To analysis red blood cell content in the bone marrow, total bone marrow cells were flushed out of femora and tibiae using PBS. Single-cell suspensions were stained in the presence of Fc receptor-blocking antibodies (mouse CD16/32, clone 93) using the following antibodies (BioLegend): CD19-FITC (clone 1D3/CD19, 152403), B220-PerCP (clone RA3-6B2, 103233), CD93-PE (clone AA4.1, 136503), CD23-APC (clone B3B4, 101619), CD21-Pacific Blue (clone 7E9, 123413), Ter119-FITC (clone TER-119, 116205) and CD45-APC-Cy7 (clone 30-F11, 103115). Cell viability was measured using Zombie Yellow Fixable Viability kit (423103) or DAPI. Flow cytometry data were acquired on the NovoCyte flow cytometer (Acea Biosciences/Agilent Technologies) using NovoExpress (version 1.3.0) and analyzed using FlowJo (BD).

### RNA-seq data analysis

Sequencing read files were processed to generate UMI[70] count matrices using the Python toolkit from the bcbio-nextgen project version 1.1.5 (https://bcbio-nextgen.readthedocs.io/en/latest/). In brief, reads were aligned to the mouse mm10 transcriptome with RapMap[71]. Quality-control metrics were compiled with a combination of FastQC (https://www.bioinformatics.babraham.ac.uk/projects/fastqc/), Qualimap and MultiQC (https://github.com/ewels/MultiQC/)[72,73]. Samples were demultiplexed using barcodes stored in read 1 (first 6 bases), and raw UMI count matrices were computed using UMIs stored in read 1 (bases 7 to 16; https://github.com/vals/umis/).

Differential expression analysis was done using custom scripts in R version 4.2.0 (https://www.r-project.org/). Raw count matrices were filtered to keep genes with at least 20 counts per million or five UMIs in two samples and normalized across samples using the calcNormFactor function in edgeR[74]. We identified genes with at least a twofold expression difference and indicated Benjamini and Hochberg FDR-adjusted $P$ value and fold expression difference by comparing treated tissues and matching control tissues using limma. Data analysis was also performed with existing packages, including tidyverse (version 2.0.0), data.table (version 1.14.8), cmapR (version 1.8.0), RColorBrewer (version 1.1–3), enrichR (version 3.1), ggrepel (version 0.9.3), patchwork (version 1.1.2), cowplot (version 1.1.0), glue (version 1.4.2), fs (version 1.3.2) and Matrix (version 1.2–18).

To assess the expression profiles of known sepsis biomarkers, we used a set of 258 genes reported as sepsis biomarkers by others[24]. We defined the absolute average log$_2$ fold change of these 258 genes within each RNA-seq profile as the sepsis biomarker score.

Heat maps for RNA-seq data display the indicated numbers of transcripts, and color intensities are determined by log$_2$ fold-change value for each heat map. The rows of each heat map were ordered by $k$-means clustering of log$_2$ fold-change values in R or Morpheus (https://software.broadinstitute.org/morpheus/). All heat maps were generated using ComplexHeatmap (version 2.12.1; https://github.com/jokergoo/ComplexHeatmap/) and circlize (version 0.4.15; https://github.com/jokergoo/circlize/) packages in R[75,76].

## Statistical modeling of cytokine pairwise effects on tissue gene expression

To assess the extent to which pairwise administration of cytokines (that is, TNF plus IL-18, IFN-γ or IL-1β) resulted in nonadditive changes (that is, synergistic or antagonistic interactions) in gene expression levels across tissues in mice, we developed an interaction scoring method based on a linear modeling method adapted from previous work[77], using custom scripts in R (https://www.r-project.org/).

First, raw, tissue RNA-seq count matrices were normalized across samples using the calcNormFactor function in edgeR[74] and subsequently filtered to keep genes with at least 15 counts per million in two samples. Data were log-transformed and a linear model was fit using the limma package[78]. We then computed the following contrasts for each pair (AB) of interest and its component singles (A, B) and unstimulated control mice:

$$\text{Single 1 effect} = A - \text{control},$$

$$\text{Single 2 effect} = B - \text{control},$$

$$\text{Pair effect} = AB - \text{control},$$

$$\text{Additive effect} = [A - \text{control}] + [B - \text{control}], \text{ and}$$

$$\text{Interaction effect} = (AB - \text{control}) - [(A - \text{control}) + (B - \text{control})].$$

Where, 'pair effect' is equivalent to the observed gene expression value for a given pair, while 'additive effect' is equivalent to the predicted gene expression value for that pair if it is assumed to be equal to the sum of the component singles. The 'interaction effect' is equal to the difference between these two values and is used as the score for assessing nonadditive interactions.

We identified genes with significantly different expression within each contrast and across all contrasts using a Benjamini and Hochberg correction for multiple-hypothesis testing and an FDR of 0.1. We next classified each gene, for each organ and pair treatment, as 'synergistic', 'antagonistic' or 'additive', depending on its score and the gene expression values of the pair and its component singles. Genes without >0.5 absolute difference in $\log_2$ fold change in at least two of the three experimentally measured treatment conditions for a given pair and organ (single 1 effect, single 2 effect, pair effect) were considered to have roughly the same expression across all samples and were excluded from further classification to avoid classifying genes with very high or low baseline expression in the singles (and, therefore, very high or low predicted additive effects but no additional increase or decrease in gene expression at the pair level) as synergistic or antagonistic.

Using the standard deviation for each contrast determined via limma, we calculated an error value, E, for each gene, as the average of the standard deviations for all experimentally measured samples for that gene. Where the score was $>2 \times E$, and the score was significant (FDR < 0.1) OR score > 1 (>2-fold difference between predicted and observed gene expression values), the gene was classified as synergistic. The gene was only classified as significantly synergistic if the score was determined to be significant at the chosen FDR (0.1). Following the same logic, if the score $< -2 \times E$ and score significant (FDR < 0.1) OR score < −1, the gene was classified as antagonistic in a particular pair and organ. Again, only if the score was determined to be significant at the chosen FDR (0.1), was the gene classified as significantly antagonistic.

The total number of DEGs was calculated by totaling any gene that showed significant differential expression (FDR < 0.1) in single 1, single 2 or pair, compared to control. The percentage of all DEGs for a given pair and organ that were classified as synergistic, additive or antagonistic was then calculated.

## Public RNA-seq data

To compare the expression profile of bacterial sepsis (this study) with that of viral sepsis induced using tissues from mice infected with a lethal dose of vaccinia virus strain Western Reserve[16], we used our previously published bulk RNA-seq data (GSE87633).

## Statistics

Statistical analyses were performed by R, using limma, or one-way ANOVA with Tukey–Kramer test. Data collection and analysis were not performed blind to the conditions of the experiments. No statistical methods were used to predetermine sample sizes, but our sample sizes are similar to those reported in previous publications[16]. Data distribution was assumed to be normal, but this was not formally tested. For experiments that require treatments, age-matched and sex-matched animals were randomly assigned into each group.

## Reporting summary

Further information on research design is available in the Nature Portfolio Reporting Summary linked to this article.

## Data availability

The sequencing data generated during this study have been deposited in the Gene Expression Omnibus under accession number GSE224146. Preprocessed datasets are available at https://doi.org/10.5281/zenodo.10158368.

## Code availability

All scripts are publicly available at https://doi.org/10.5281/zenodo.10158368.

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

## Acknowledgements

We thank members of the N.C. laboratory for valuable discussions; UChicago core facilities and services: the Single Cell Immunophenotyping Core (M. Abasiyanik, H.-N. Shim), the Animal Resources Center (A. Solanki), the Research Computing Center, the Integrated Light Microscopy Core and the Human Tissue Resource Center; the In Vivo Animal Core at the University of Michigan; and SciStories for help with artwork. M.T. was supported by the Astellas Foundation for Research on Metabolic Disorders and K.J. by the National Science Foundation Graduate Research Fellowship under grant no. 2140001. N.C. was supported by National Institutes of Health grants DP2-AI145100 and U01-AI160418, the CZI grant DAF2020-217464 and a grant from the Chan Zuckerberg Initiative DAF (https://doi.org/10.37921/767230ofotux), an advised fund of Silicon Valley Community Foundation (funder https://doi.org/10.13039/100014989), the University of Chicago Medicine Comprehensive Cancer Center (UCCCC) Janet D. Rowley Discovery Fund, the University of Chicago Center for Interdisciplinary Study of Inflammatory Intestinal Disorders (C-IID; P30 DK42086) and funds from the Chicago Immunoengineering Innovation Center and the Pritzker School of Molecular Engineering at the University of Chicago.

## Author contributions

Conceptualization and methodology, M.T. and N.C.; investigation, M.T., G.R., M.P., D.C., S.P., K.C., T.U. and N.C.; cell-type abundance prediction analysis, A.P., M.T. and N.C.; topic modeling analysis, P.C. and M.S.; gene expression interaction scoring, K.J., M.S. and N.C.; formal analysis, M.T., D.C., A.G. and N.C.; resources, T.K. and N.C.; writing—original draft, M.T. and N.C.; writing—review and editing, all authors; funding acquisition and supervision, N.C.

## Competing interests

A.P. is the founder and CEO of Combinatics. All other authors declare no competing interests.

## Additional information

**Extended data** is available for this paper at https://doi.org/10.1038/s41590-023-01722-8.

**Correspondence and requests for materials** should be addressed to Nicolas Chevrier.

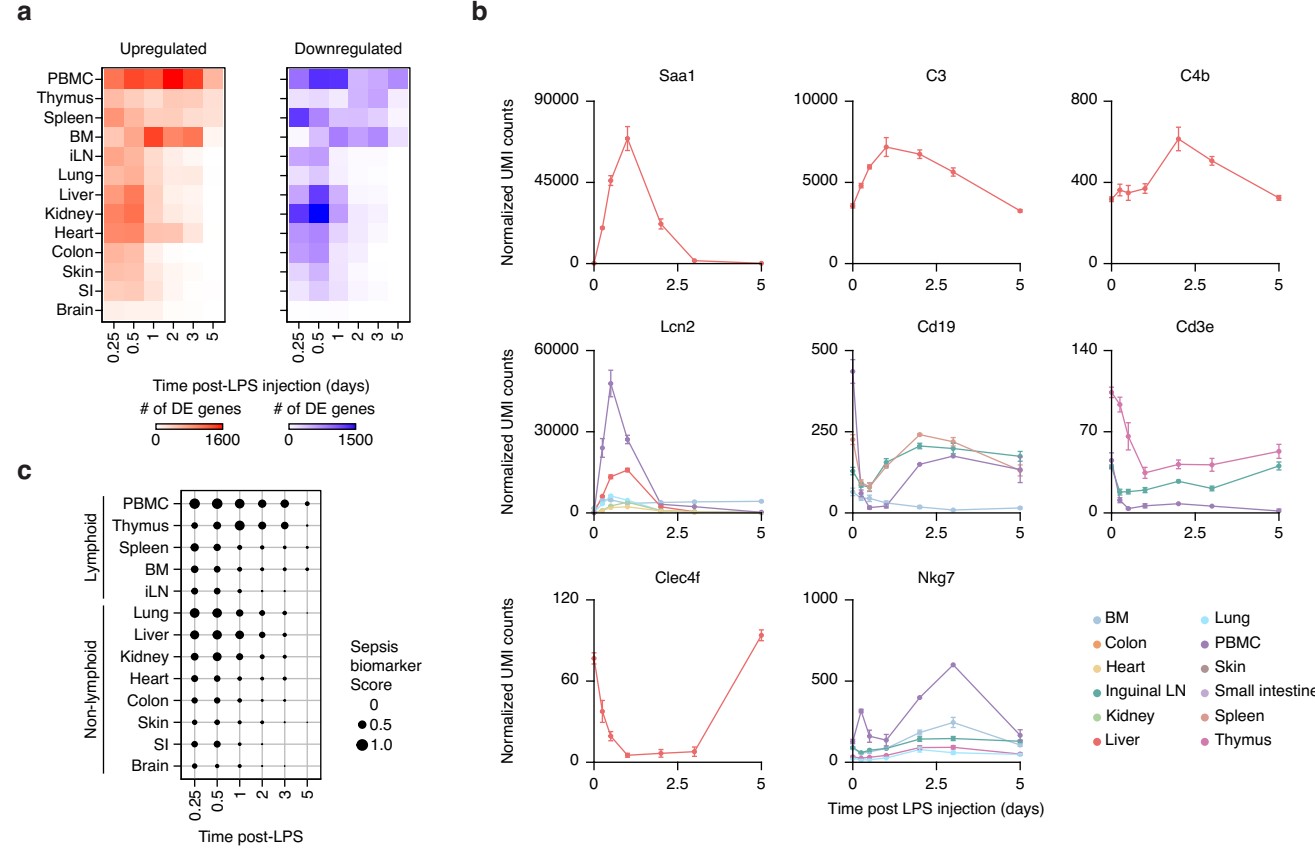

**Extended Data Fig. 1 | Multi-Tissue Expression Analysis of LPS-Induced Sepsis. a**, Heatmaps showing the total numbers of up- (left) and down-regulated (right) genes across time post-LPS (columns) for each organ (rows). **b**, Normalized counts for indicated genes and organs (color). BM, bone marrow; iLN, inguinal lymph node; PBMCs, peripheral blood mononuclear cells; SI, small intestine. n = 2 biologically independent samples for PBMC 2 days, or 3 days; n = 3 for BM 5 days, colon 0.25 day, iLN. 2 days, liver 1 day, lung 3 days; n = 4 for other groups. **c**, Expression of sepsis biomarker genes (score) in each organ (rows) at each time point post-sublethal LPS injection (columns). Rows are ordered from top to bottom by high to low scores and by lymphoid and non-lymphoid tissues.

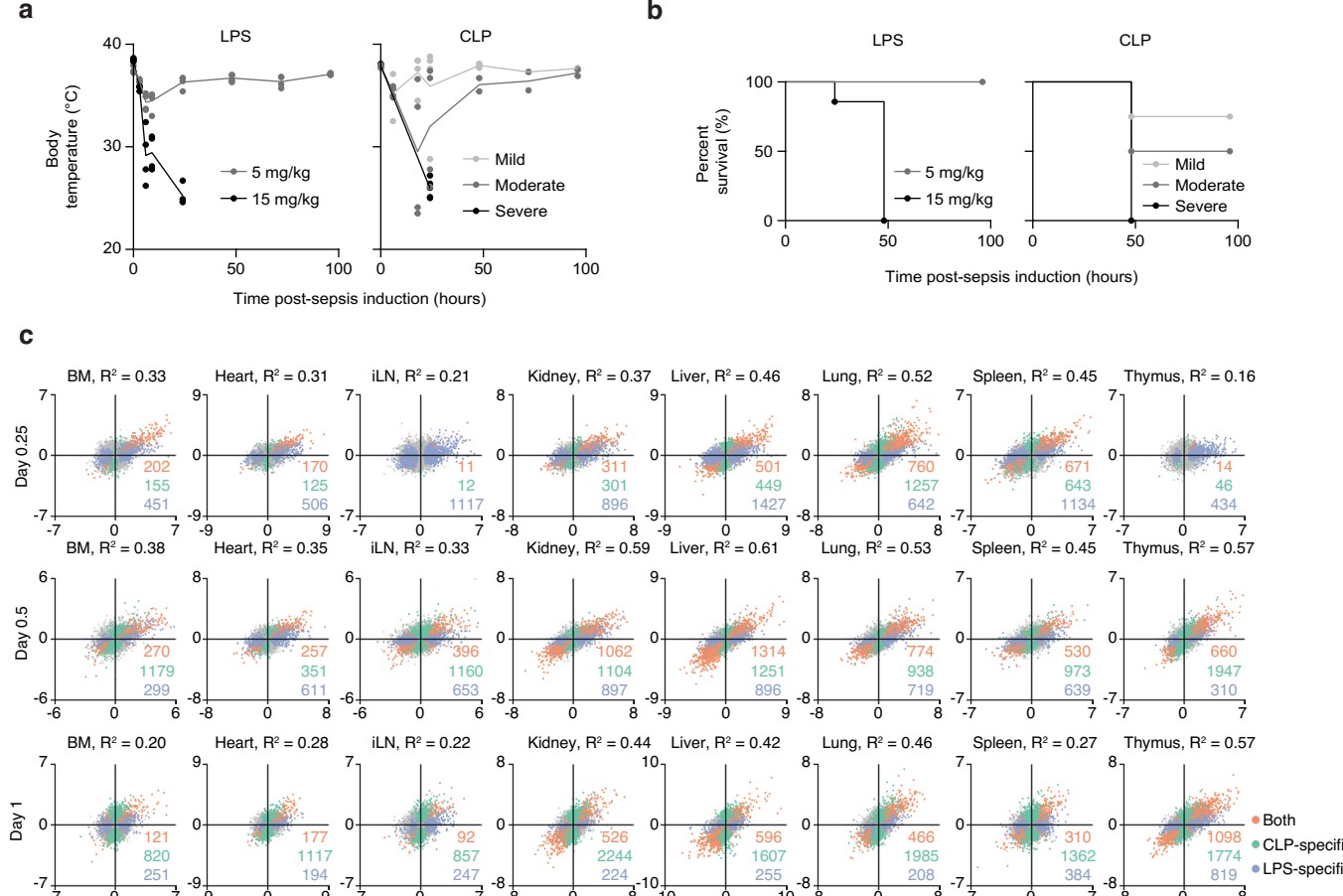

**Extended Data Fig. 2 | Comparative, Multi-Tissue Expression Analysis across Bacterial Sepsis Models. a**, **b**, Measurements of rectal temperature (a) and survival (b) after LPS injection at sublethal (5 mg/kg) or lethal (15 mg/kg) doses (left, n = 4 biologically independent samples), or cecal ligation and puncture (CLP) surgeries leading to severe sepsis and sham control (Methods) (right, n = 5 biologically independent samples). **c**, Dot plots showing log2 fold-change in gene expression in tissues collected at 0.5 day after sublethal LPS injection (x-axis) or severe CLP (y-axis) relative to matching organs from untreated mice for LPS and mice after sham surgery for CLP. Colored dots represent genes regulated in LPS only with FDR < 0.01 and absolute fold change > 2 (LPS-specific; blue), CLP only with FDR < 0.1 (CLP-specific; green), and LPS and CLP (both; orange). BM, bone marrow; iLN, inguinal lymph node.

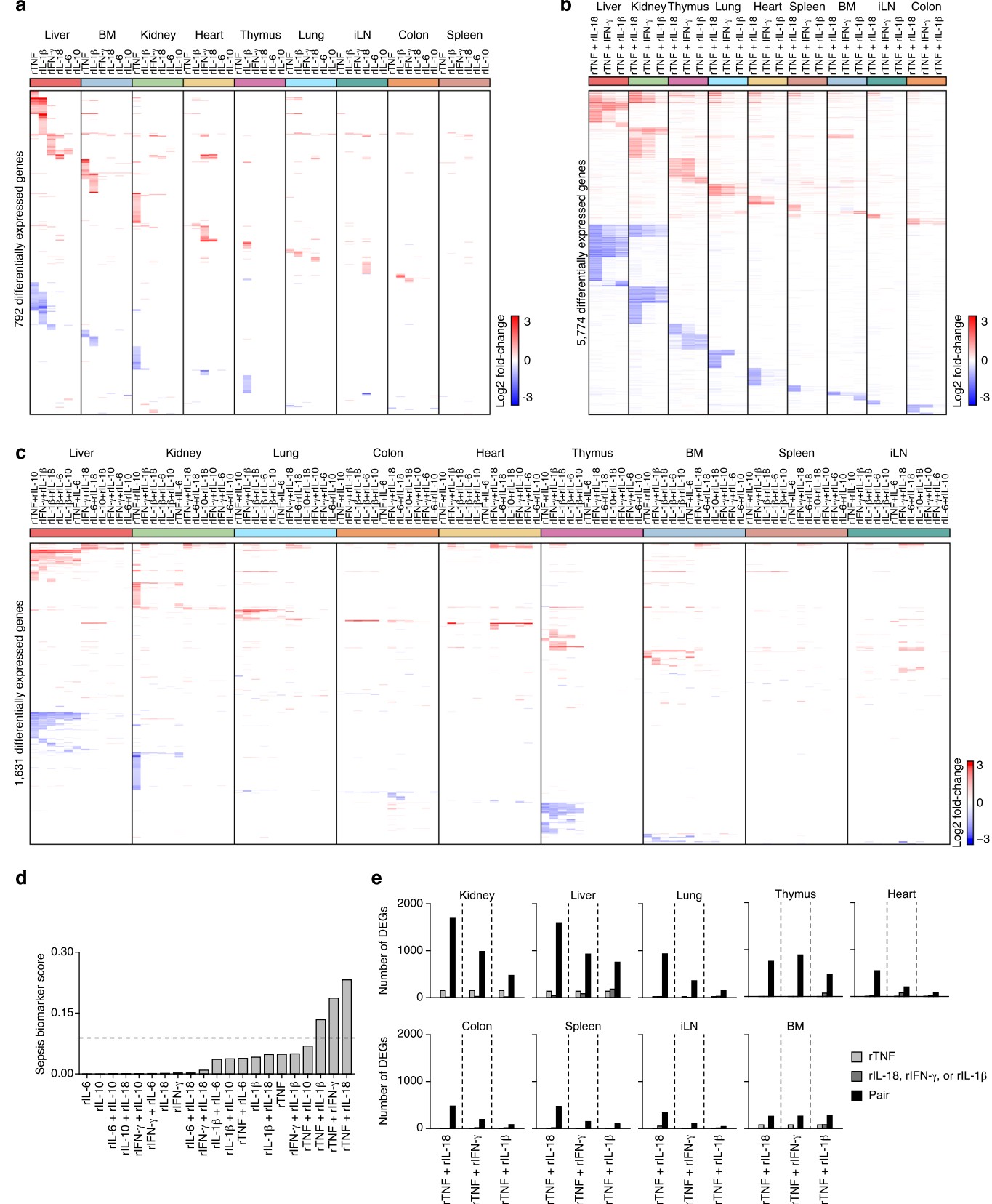

**Extended Data Fig. 3 | See next page for caption.**

**Extended Data Fig. 3 | Recombinant Cytokines Injected Alone or in Pairwise Combinations Impact Tissue Transcriptional States. a**–**c**, Heatmaps of differentially expressed genes (rows) from whole-tissue mRNA profiles ordered by k-means clustering and organ types (top, colors) at 16 hours after injection with indicated recombinant cytokines used alone (a), in three pairwise combinations (b), or in other pairwise combinations (c). Values are log2 fold-changes relative to matching organs from untreated, control mice. Statistical analyses were performed with limma (FDR-adjusted p-value < 0.05, absolute fold change > 2). BM, bone marrow; iLN, inguinal lymph node. **d**, Sepsis biomarker score average (scaled by condition) across all 9 organs profiled in a, b, and c in indicated recombinant cytokine conditions (x axis). Dashed line indicates one standard deviation. Sepsis biomarker score is the average, absolute log2 fold-change values for all 258 genes identified as sepsis biomarker in the literature. **e**, Numbers of genes (y axis) regulated in each tissue type by indicated cytokine pairs and composite singles.

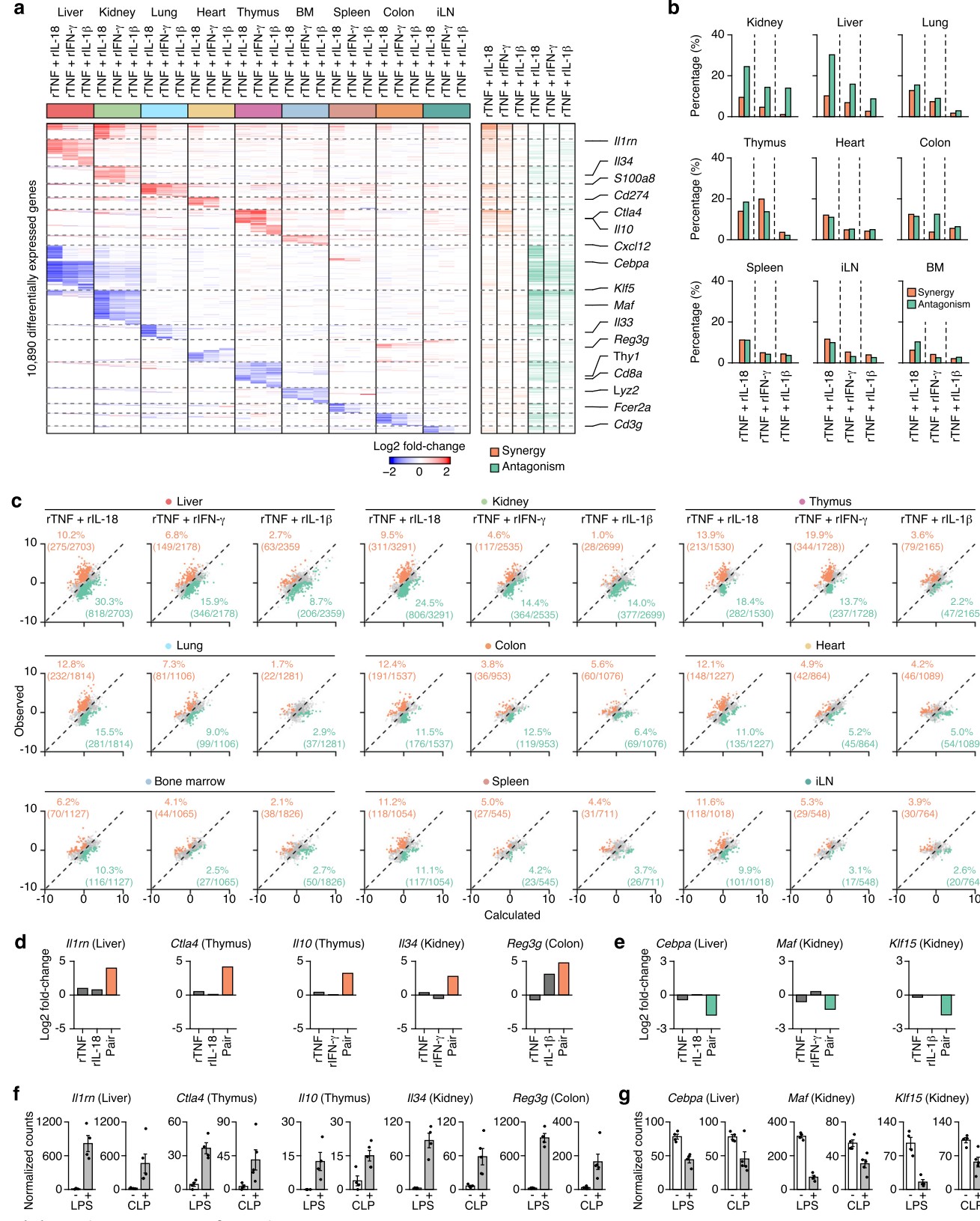

**Extended Data Fig. 4 | See next page for caption.**

**Extended Data Fig. 4 | The Cytokine PairsComposed of TNF plus IL-18, IFN-γ, or IL-1β Yield Nonlinear Effects on Tissue Transcriptional States. a**, Heatmap (left) of differentially expressed genes (rows) from whole-tissue mRNA profiles ordered by k-means clustering and organ types (top, colors) at 16 hours after injection of indicated recombinant cytokine pairs. Values are log2 fold-changes relative to matching, untreated organs. Statistical analyses were performed with limma (FDR-adjusted p-value < 0.1). Genes synergistically or antagonistically regulated by the indicated recombinant cytokine pairs relative to matching single cytokines in at least of one of the 9 tissues profiled are indicated in orange and green, respectively (right). BM, bone marrow; iLN, inguinal lymph node. **b**, Percentages (y axis) of differentially expressed genes in each tissue type displaying synergistic (orange) or antagonistic (green) in indicated cytokine pairs relative to matching single cytokines. **c**, Dot plots of the observed (y axis) and calculated (x axis) pairwise cytokine interaction effects relative to matching single cytokines on DEGs (dots) in indicated organs (top). Percentages and absolute counts of DEGs classified as synergistic (orange) or antagonistic (green) upon pairwise cytokine injection relative to matching singles. **d**–**g**, Log2 fold-changes (d-e) or normalized counts (f-g) for indicated tissues and genes with nonlinear regulation (orange, synergistic; green, antagonistic) in mice injected with indicated cytokines (d-e) or upon LPS or CLP sepsis (f-g). Error bars, SEM (n = 3 biologically independent samples for liver: rTNF plus rIL-18, thymus: rTNF, rTNF plus rIL-18, rTNF plus IFN-γ and colon: rTNF; n = 5 for CLP; n = 4 for other groups).

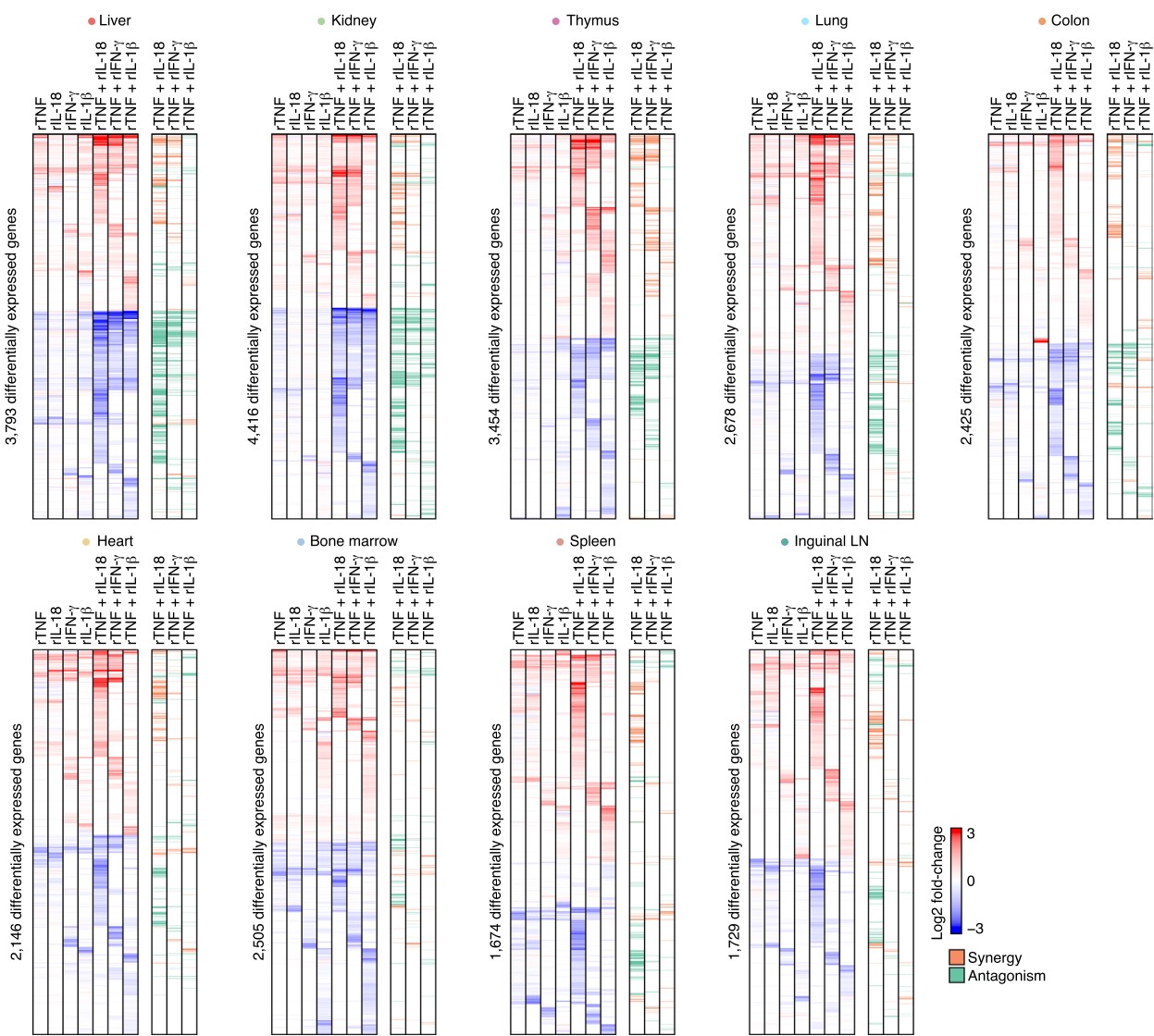

**Extended Data Fig. 5 | The Cytokine Pairs Composed of TNF plus IL-18, IFN-γ, or IL-1β Yield Nonlinear Interactions at the Gene Expression Level across Organs.** Heatmaps of differentially expressed genes (rows) from whole-tissue mRNA profiles for each indicated organ ordered by k-means clustering at 16 hours after injection of indicated recombinant cytokines. Values are log2 fold-changes relative to matching, untreated organs. Statistical analyses were performed with limma (FDR-adjusted p-value < 0.05; absolute fold change > 2). Genes synergistically or antagonistically regulated by the indicated recombinant cytokine pairs relative to matching single cytokines are indicated in orange and green, respectively. LN, lymph node.

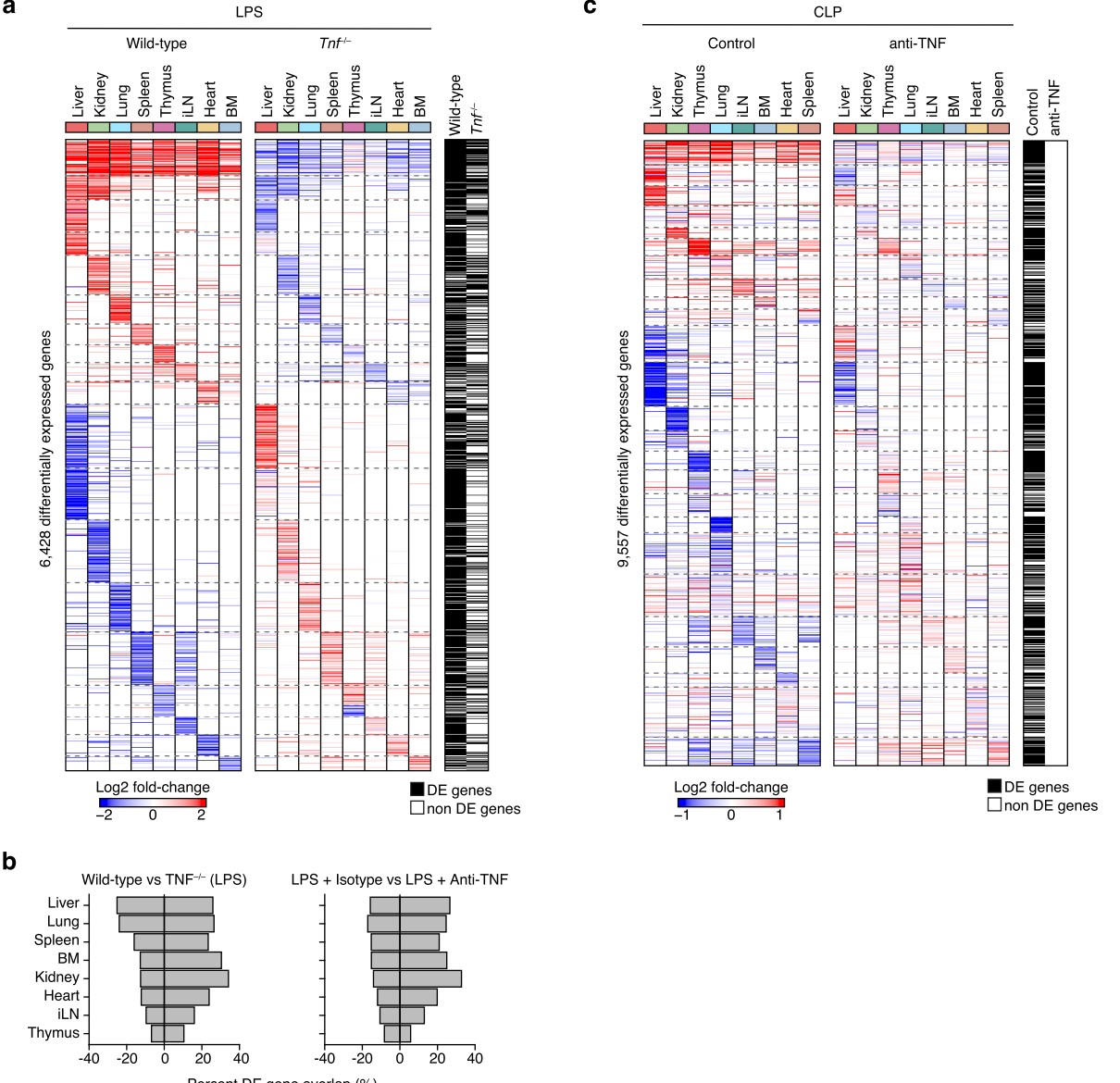

**Extended Data Fig. 6 | The Effect of TNF on the Transcriptional Responses of Organs across Bacterial Sepsis. a**, Heatmaps of differentially expressed genes (rows) from whole-tissue mRNA profiles ordered by k-means clustering and organ types (top, colors) at 0.5 day after sublethal LPS injection in wild-type (left) or *Tnf*⁻/⁻ mice (right). Values are log2 fold-changes relative to matching organs from untreated mice for wild-type, or LPS-treated wild-type mice for *Tnf*⁻/⁻. Statistical analyses were performed with limma (FDR-adjusted p-value < 0.01, absolute fold change > 2, or FDR-adjusted p-value < 0.05, absolute fold change > 2). Shown are all the genes found to be differentially regulated in at least one of the indicated conditions (row annotations in black). BM, bone marrow; iLN, inguinal lymph node. **b**, Percentages (x axis) of genes differentially expressed in tissues (rows) upon sublethal LPS injection in *Tnf*⁻/⁻ mice (left) or mice treated

with anti-TNF antibodies (1 h prior to LPS; right) that match the genes regulated by LPS in wild-type mice. **c**, Heatmaps of differentially expressed genes (rows) from whole-tissue mRNA profiles ordered by k-means clustering and organ types (top, colors) at 0.5 day after cecal ligation and puncture (CLP) with pre-treatment with anti-TNF (right) or isotype control (left) antibodies. Values are log2 fold-changes relative to matching organs from sham operated mice for CLP with isotype control antibodies, or CLP operated mice for CLP with anti-TNF antibodies. Statistical analyses were performed with limma (FDR-adjusted p-value < 0.1, or FDR-adjusted p-value < 0.1). Shown are all the genes found to be differentially regulated in at least one of the indicated conditions (row annotations in black).

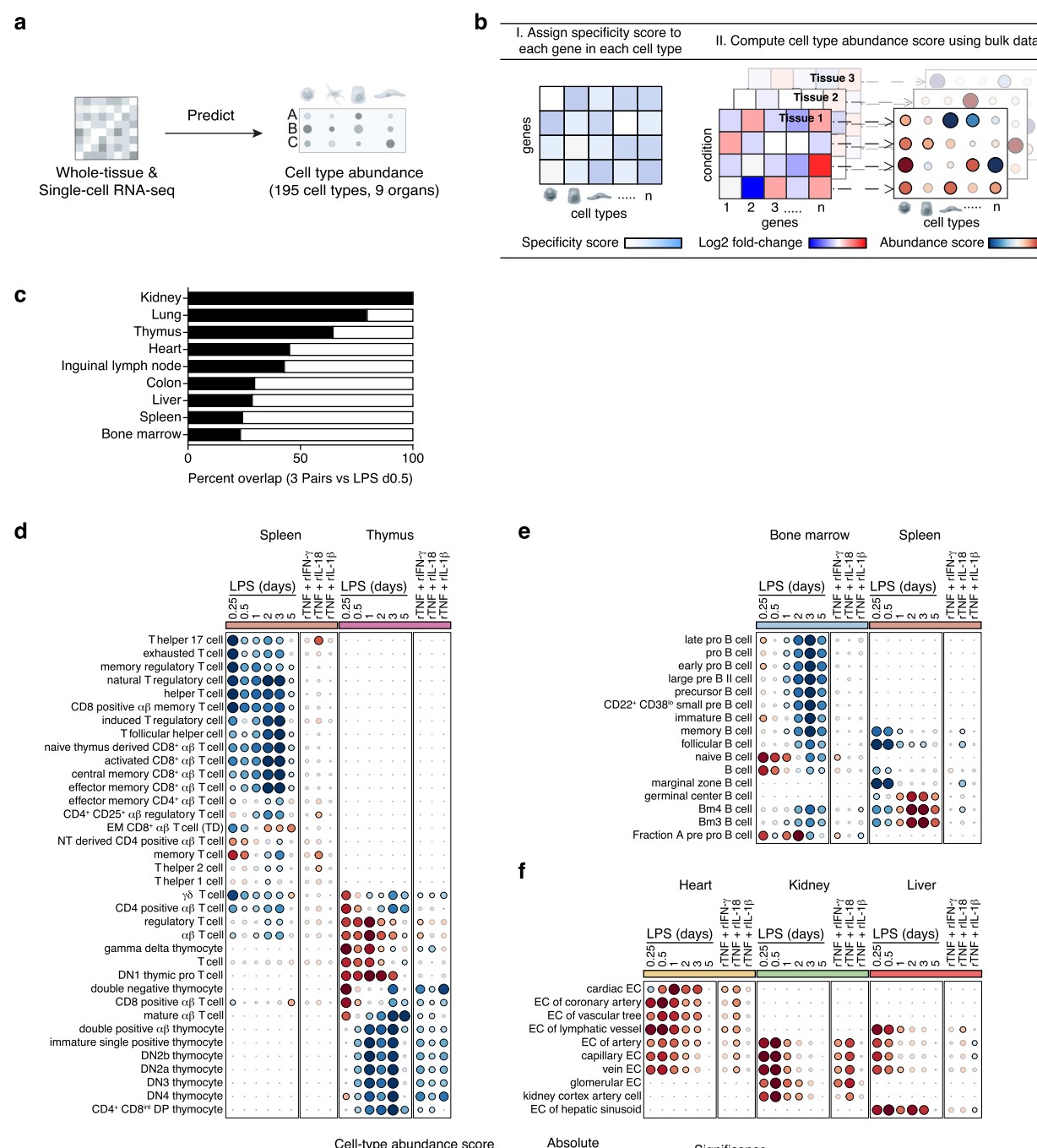

**Extended Data Fig. 7 | The Pairwise Effects of TNF plus IL-18, IFN-γ, or IL-1β Lead to Well-Known Sepsis Effects on Cells from Lymphoid and Non-Lymphoid Tissues. a**, Schematic overview of the analytical workflow to predict changes in cell type abundances during sepsis or upon recombinant cytokine injections from bulk, whole-tissue gene expression data. **b**, Schematic overview of the method to computationally estimate the relative abundance of cell types in organs from treated (LPS, recombinant cytokines) versus untreated, control mice by combining cell type-specific gene sets and whole-tissue gene expression measurements. **c**, Percentages (black bars; x axis) of the effects of LPS on cell type abundance scores across tissues (y axis) mirrored by at least one of the three cytokine pairs tested: TNF plus IL-18, IFN-γ, or IL-1β. **d**–**f**, Cell type abundance scores computed for indicated cell types (rows) and tissues (colors; top) upon injection of a sublethal dose of LPS in wild-type (left) or injection of indicated recombinant cytokine pairs (right) (columns). Black borders indicate significance (z-score > 1). EM, effector memory; TD, terminally differentiated; NT, naive thymus; DP, double positive; EC, epithelial cell.

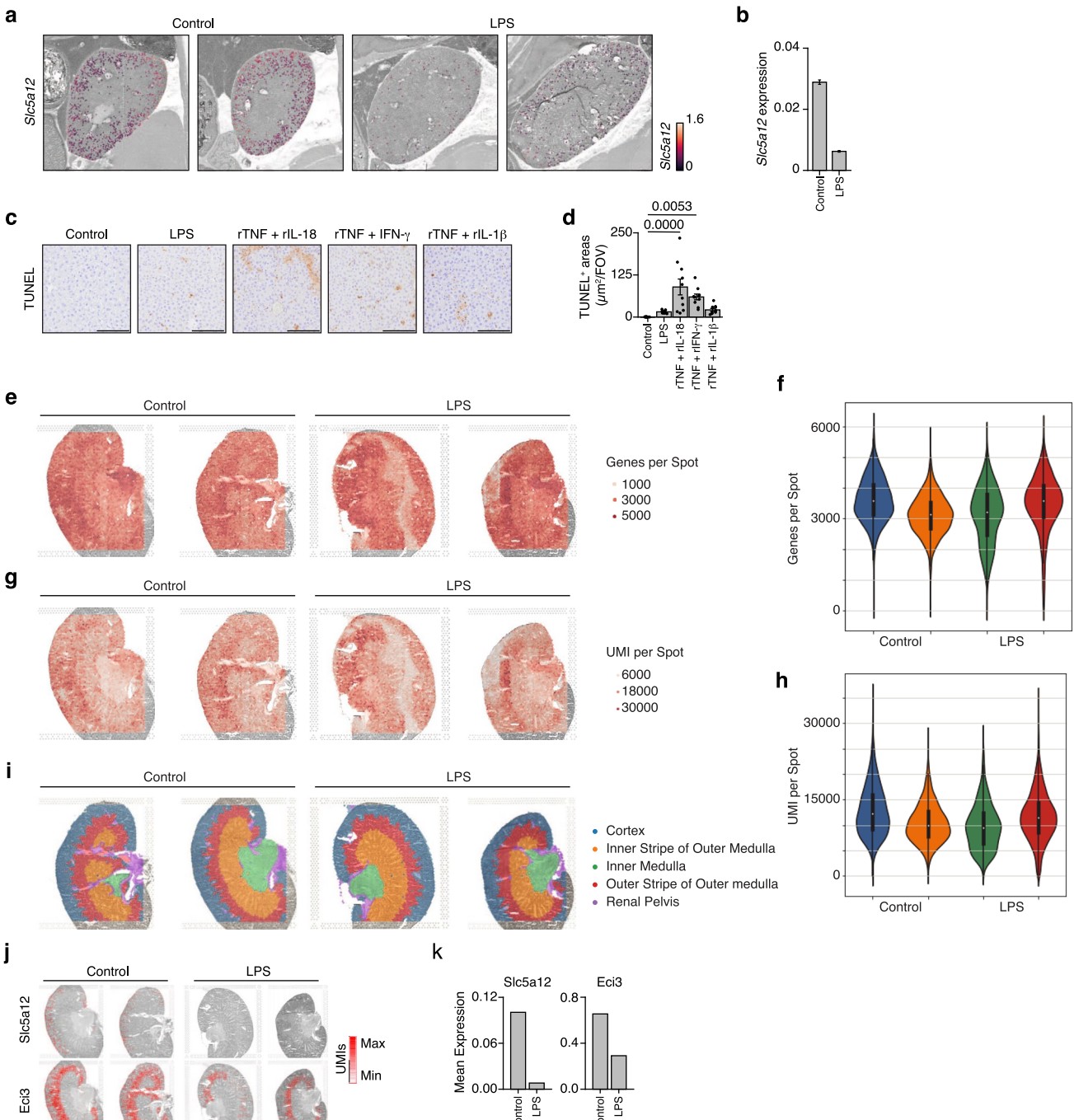

**Extended Data Fig. 8 | Experimental Validation of Changes in Cell Type Abundance Scores Computed from Whole-Tissue Gene Expression Profiles in Kidney and Liver. a**, **b**, Whole-mouse spatial transcriptomics (ST) data (a) from control and LPS conditions (columns) were magnified to only show kidney tissues. Slc5a12 normalized expression was overlaid as cell type markers on a greyscale H&E image. Bar plot (b) of average expression of indicated genes across all ST array spots covering indicated tissues. Error bars, SEM (n > 10, the number of ST array spots covering kidney). **c**, **d**, Images (40X magnification; c) from TUNEL staining in liver from mice injected with LPS, indicated cytokines, or left untreated as controls. Bar graph (d) shows quantifications of TUNEL+ areas (μm²) per field of view. Scale bars, 100 μm; Error bar, SEM (n = 10 independent field of view). **e**–**k**, Grey-scale H&E images from mouse kidney sections (n = 2) from PBS- (control) or LPS-treated mice processed for commercial spatial transcriptomics platform and overlaid with the numbers of genes (e) or UMIs (g) detected per spot, or with spatial clusters annotated with known kidney histological regions (i). Violin plots show the matching distributions of the numbers of genes (f) and UMIs (h) per spot. Overlayed box plots (f, h) show the median, 25th and 75th percentiles. Spatial gene expression analysis (j) of indicated genes (rows) from control or LPS-treated mice (columns) overlaid on grey-scale H&E images from mouse kidney sections. Bar graphs (k) show the mean expression of each gene (top) across spatial transcriptomics spots and replicate sections (n = 2 biologically independent samples).

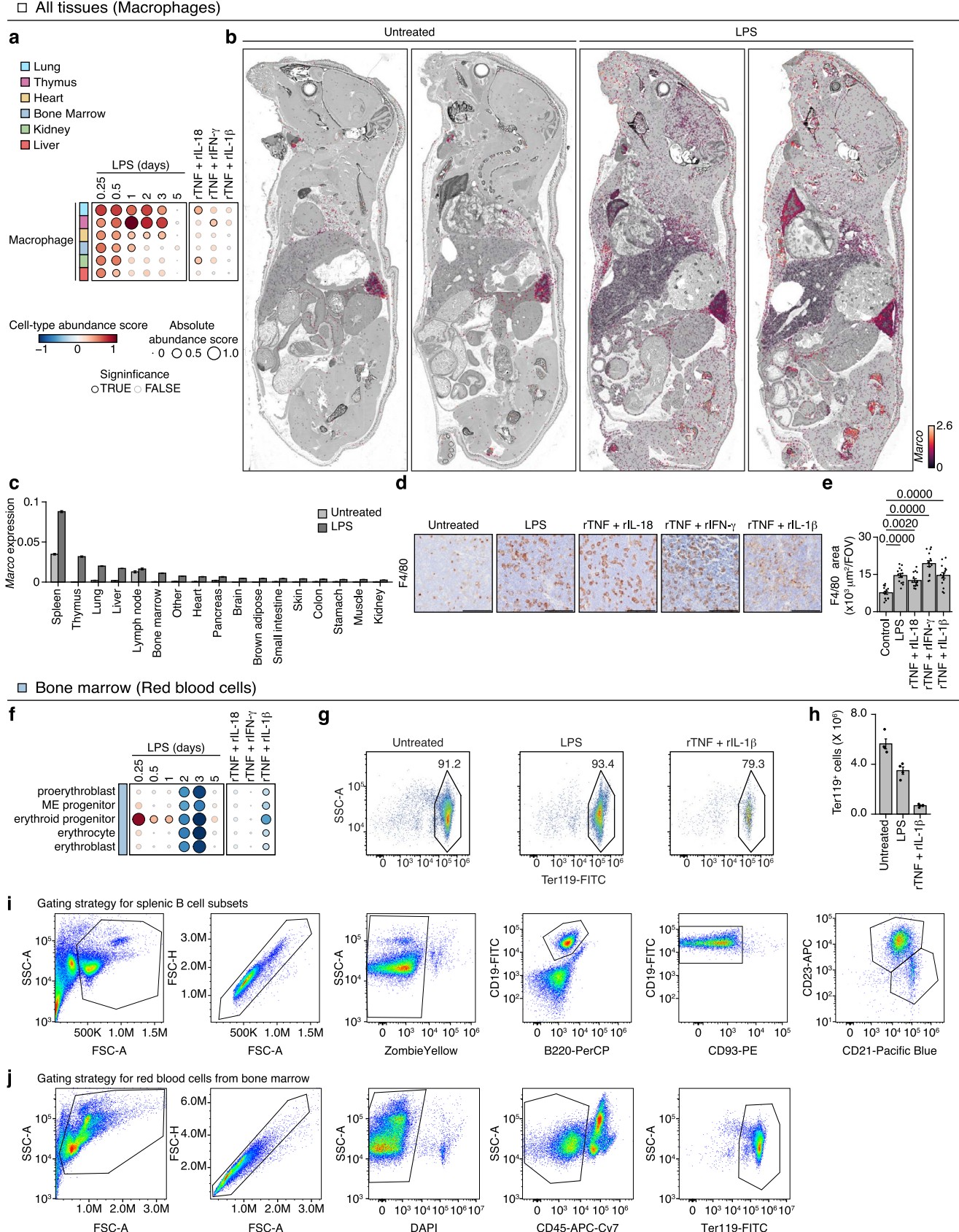

**Extended Data Fig. 9 | See next page for caption.**

**Extended Data Fig. 9 | Experimental Validation of Changes in Cell Type Abundance Scores Computed from Whole-Tissue Gene Expression Profiles in Lung, Spleen, and Thymus. a**, Cell type abundance scores computed for macrophages upon injection of a sublethal dose of LPS in wild-type (left) or injected with indicated recombinant cytokine pairs (right) (columns). **b**, **c**, Whole-mouse spatial transcriptomics (ST) analysis (b) of Marco mRNA levels overlaid on a greyscale H&E staining. Shown are whole-mount sections and ST data from 5-weeks old mice injected with a sublethal dose of LPS (5 mg/kg) or left untreated as control. Bar plot (c) of average expression of Marco across all ST array spots covering indicated tissues. Error bar, SEM (n > 10, the number of ST array spots covering indicated tissues). **d**, **e**, Images (40X magnification;

d) from F4/80 immunohistochemistry in thymus tissues from mice injected with LPS, indicated cytokines, or left untreated as controls. Bar graph (e) shows quantifications of F4/80$^+$ areas per field of view (FOV) from (**d**). Scale bars, 100 μm; Error bar, SEM (n = 15 independent field of view). **f**, Cell type abundance scores computed for red blood cells upon injection of a sublethal dose of LPS in wild-type (left) or injected with indicated recombinant cytokine pairs (right) (columns). **g**, **h**, Flow cytometry analysis (g) of bone marrow erythrocytes from mice injected with a sublethal dose of LPS or indicated cytokines. Bar graph (h) shows quantifications in absolute count per tissue (n = 4 biologically independent samples). **i**, **j**, Flow cytometry plots of gating strategy used for splenic B cells (i) and erythrocytes (j).

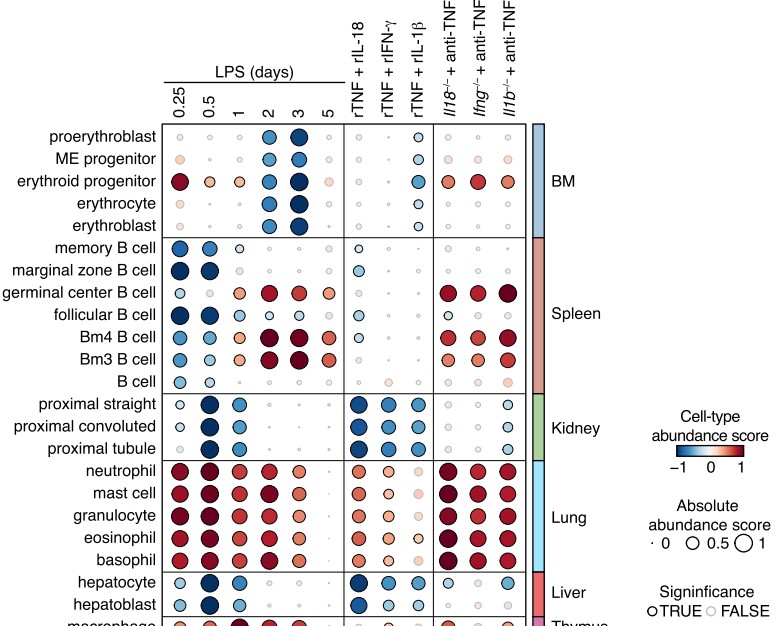

**Extended Data Fig. 10 | The Effects of perturbation on Cell Type Abundance Scores Computed from Whole-Tissue Gene Expression Profiles.** Cell type abundance scores computed for indicated cell types (row) upon injection of a sublethal dose of LPS in wild-type (left) or indicated knockout animals pretreated with indicated neutralizing antibodies (right) or injected with indicated recombinant cytokine pairs (center) (columns). BM, bone marrow.

# Reporting Summary

## Statistics

For all statistical analyses, confirm that the following items are present in the figure legend, table legend, main text, or Methods section.

| n/a | Confirmed | |
|---|---|---|
| ☐ | ☒ | The exact sample size (*n*) for each experimental group/condition, given as a discrete number and unit of measurement |
| ☐ | ☒ | A statement on whether measurements were taken from distinct samples or whether the same sample was measured repeatedly |
| ☐ | ☒ | The statistical test(s) used AND whether they are one- or two-sided *Only common tests should be described solely by name; describe more complex techniques in the Methods section.* |
| ☐ | ☒ | A description of all covariates tested |
| ☐ | ☒ | A description of any assumptions or corrections, such as tests of normality and adjustment for multiple comparisons |
| ☐ | ☒ | A full description of the statistical parameters including central tendency (e.g. means) or other basic estimates (e.g. regression coefficient) AND variation (e.g. standard deviation) or associated estimates of uncertainty (e.g. confidence intervals) |
| ☐ | ☒ | For null hypothesis testing, the test statistic (e.g. *F*, *t*, *r*) with confidence intervals, effect sizes, degrees of freedom and *P* value noted *Give P values as exact values whenever suitable.* |
| ☒ | ☐ | For Bayesian analysis, information on the choice of priors and Markov chain Monte Carlo settings |
| ☐ | ☒ | For hierarchical and complex designs, identification of the appropriate level for tests and full reporting of outcomes |
| ☒ | ☐ | Estimates of effect sizes (e.g. Cohen's *d*, Pearson's *r*), indicating how they were calculated |

*Our web collection on statistics for biologists contains articles on many of the points above.*

## Software and code

Policy information about availability of computer code

| Data collection | Flow cytometry data and LEGENDplex data were collected with NovoExpress (version 1.3.0) and LEGENDplex software v8. Bulk and spatial RNA-seq data collection and immunohistochemistry image acquisition are detailed in the methods section of the paper. |
|---|---|
| Data analysis | The following software packages were used for data analysis: Flow cytometry data were analyzed with FlowJo (Version 10.6.2) software. Raw sequencing data were processed using bcbio-nextgen project (version 1.1.5) or STARsolo (version 2.7.10a). Integrated data analysis was performed in R (version 4.2.0 and 3.5.1) or Python (version 3.8.5), based on the packages tidyverse (version 2.0.0), fastTopics (version 0.6-142), data.table (version 1.14.8), cmapR (version 1.8.0), ComplexHeatmap (version 2.12.1), circlize (version 0.4.15), RColorBrewer (version 1.1-3), enrichR (version 3.1), ggrepel (version 0.9.3), patchwork (version 1.1.2), edgeR (version 3.24.3), limma (version 3.38.3), cowplot (version 1.1.0), glue (version 1.4.2), fs (version 1.3.2), Matrix (version 1.2-18), Scanpy (version 1.9.1), scikit-image (version 1.1.3), sckit-learn (version 0.24.2), CARD (version 1.0.0), seaborn (version 0.11.2). ImageJ (version 2.8.0) was used for image analysis. The Space Ranger software was used to process, align, and summarize the FASTQ files. Adobe Illustrator (version 26.2.1) was used for data visualization and figure preparation. Source code is available at 10.5281/zenodo.10158368. |

For manuscripts utilizing custom algorithms or software that are central to the research but not yet described in published literature, software must be made available to editors and reviewers. We strongly encourage code deposition in a community repository (e.g. GitHub). See the Nature Portfolio guidelines for submitting code & software for further information.

## Data

Policy information about <u>availability of data</u>

All manuscripts must include a <u>data availability statement</u>. This statement should provide the following information, where applicable:
- Accession codes, unique identifiers, or web links for publicly available datasets
- A description of any restrictions on data availability
- For clinical datasets or third party data, please ensure that the statement adheres to our <u>policy</u>

> Raw and processed sequencing data are available from the NCBI Gene Expression Omnibus (GEO) repository (accession number: GSE224146).
> Source code and preprocessed datasets are available at 10.5281/zenodo.10158368.
> We used GRCm39 (GCF_000001635.27) as mouse reference genome.

## Research involving human participants, their data, or biological material

Policy information about studies with <u>human participants or human data</u>. See also policy information about <u>sex, gender (identity/presentation), and sexual orientation</u> and <u>race, ethnicity and racism</u>.

| | |
|---|---|
| Reporting on sex and gender | N/A |
| Reporting on race, ethnicity, or other socially relevant groupings | N/A |
| Population characteristics | N/A |
| Recruitment | N/A |
| Ethics oversight | N/A |

Note that full information on the approval of the study protocol must also be provided in the manuscript.

# Field-specific reporting

Please select the one below that is the best fit for your research. If you are not sure, read the appropriate sections before making your selection.

☒ Life sciences  ☐ Behavioural & social sciences  ☐ Ecological, evolutionary & environmental sciences

For a reference copy of the document with all sections, see <u>nature.com/documents/nr-reporting-summary-flat.pdf</u>

# Life sciences study design

All studies must disclose on these points even when the disclosure is negative.

| | |
|---|---|
| Sample size | No statistical methods were used to predetermine sample sizes, but our sample sizes are similar to those reported in previous publications |
| Data exclusions | No samples were excluded, unless data acquisition quality was insufficient (e.g., low quality of RNA, or low raw read counts for RNA-seq). |
| Replication | Biological replicate samples were detailed in the accompanying figure legends or methods. The results were consistent in all independent experiments. |
| Randomization | No randomization was done as it is not relevant for allocation of mice. Mice were selected according to their genotype, age and sex matched. Mice were randomly assigned for experiments reported. Processing of samples from the various organs did not follow any particular order. |
| Blinding | Data collection and analysis were not performed blind to the conditions of the experiments. |

# Reporting for specific materials, systems and methods

We require information from authors about some types of materials, experimental systems and methods used in many studies. Here, indicate whether each material, system or method listed is relevant to your study. If you are not sure if a list item applies to your research, read the appropriate section before selecting a response.

## Materials & experimental systems

| n/a | Involved in the study |
|-----|----------------------|
| ☐ | ☒ Antibodies |
| ☒ | ☐ Eukaryotic cell lines |
| ☒ | ☐ Palaeontology and archaeology |
| ☐ | ☒ Animals and other organisms |
| ☒ | ☐ Clinical data |
| ☒ | ☐ Dual use research of concern |
| ☒ | ☐ Plants |

## Methods

| n/a | Involved in the study |
|-----|----------------------|
| ☒ | ☐ ChIP-seq |
| ☐ | ☒ Flow cytometry |
| ☒ | ☐ MRI-based neuroimaging |

# Antibodies

| | |
|---|---|
| Antibodies used | CD16/CD32 blocking antibody (clone 93, BioLegend Cat# 101, 1 µL/sample)<br>https://www.biolegend.com/ja-jp/products/purified-anti-mouse-cd16-32-antibody-190<br>CD19-FITC (clone 1D3/CD19, BioLegend Cat# 152403, 2 µL/sample)<br>https://www.biolegend.com/ja-jp/products/fitc-anti-mouse-cd19-antibody-13615<br>B220-PerCP (clone RA3-6B2, BioLegend Cat# 103233, 4 µL/sample)<br>https://www.biolegend.com/ja-jp/products/percp-anti-mouse-human-cd45r-b220-antibody-4266<br>CD93-PE (clone AA4.1, BioLegend Cat# 136503, 4 µL/sample)<br>https://www.biolegend.com/ja-jp/products/pe-anti-mouse-cd93-aa4-1-early-b-lineage-antibody-6314<br>CD23-APC (clone B3B4, BioLegend Cat# 101619, 1 µL/sample)<br>https://www.biolegend.com/ja-jp/products/apc-anti-mouse-cd23-antibody-9762<br>CD21-Pacific Blue (clone 7E9, BioLegend Cat# 123413, 1 µL/sample)<br>https://www.biolegend.com/ja-jp/products/pacific-blue-anti-mouse-cd21-cd35-cr2-cr1-antibody-4336<br>Ter119-FITC (clone TER-119, BioLegend Cat# 116205, 4 µL/sample)<br>https://www.biolegend.com/ja-jp/products/fitc-anti-mouse-ter-119-erythroid-cells-antibody-1865<br>CD45-APC-Cy7 (clone 30-F11, BioLegend Cat#103115, 2 µL/sample)<br>https://www.biolegend.com/ja-jp/products/apc-cyanine7-anti-mouse-cd45-antibody-2530<br>Zombie Yellow Fixable Viability kit (BioLegend Cat# 423103, 0.5 µL/sample)<br>https://www.biolegend.com/ja-jp/products/zombie-yellow-fixable-viability-kit-8514<br>DAPI (Biotium Cat#40043, 1 µL/sample)<br>https://biotium.com/product/dapi/<br>anti-TNF (clone XT3.11, BioXCell Cat#BE0058, 50 µg/mouse)<br>https://bioxcell.com/invivomab-anti-mouse-tnfa?<br>gad_source=1&gclid=EAIaIQobChMIh_HKo8vUggMVQhJ7Bx3SOwblEAAYASAAEgKCYfD_BwE<br>anti-IL-18 (clone YIGIF74-1G7, BioXCell Cat#BE0237, 50 µg/mouse)<br>https://bioxcell.com/invivomab-anti-mouse-il-18-be0237<br>anti-IFN-γ (clone XMG1.2, BioXCell Cat#BE0055, 50 µg/mouse)<br>https://bioxcell.com/invivomab-anti-mouse-ifng-be0055<br>anti-IL-1β (clone B122, BioXCell Cat#BE0246, 50 µg/mouse)<br>https://bioxcell.com/invivomab-anti-mouse-rat-il-1b<br>anti-Ly-6G (clone 1A8, BioLegend Cat#127602, 1:100)<br>https://www.biolegend.com/ja-jp/products/purified-anti-mouse-ly-6g-antibody-4767?GroupID=BLG7232<br>anti-F4/80 (clone BM8, BioLegend Cat#123102, 1:100)<br>https://www.biolegend.com/ja-jp/products/purified-anti-mouse-f4-80-antibody-4064 |
| Validation | All antibodies are commercially available and have been validated by manufacturers and in previous publications. The manufacturers' websites provide details regarding validations and associated reference publication (see above section). For flow cytometry, antibodies were titrated in the laboratory before use. |

# Animals and other research organisms

Policy information about studies involving animals; ARRIVE guidelines recommended for reporting animal research, and Sex and Gender in Research

| | |
|---|---|
| Laboratory animals | C57BL/6J mice (wild-type, stock 000664), B6.129S7-Ifngtm1Ts/J (Ifng KO, stock 002287), B6.129P2-Il18tm1Aki/J (Il18 KO, stock 004130), C57BL/6J-Il1bem2Lutzy/Mmjax (Il1b KO, stock 068082-JAX), and B6.129S-Tnftm1Gkl/J (Tnf KO, stock 005540) were obtained from the Jackson Laboratories. For all experiments, mice were used at 5-8 weeks of age. Animals were housed in specific pathogen-free and BSL2 conditions at The University of Chicago. Mice were on 12hr light/dark cycles with daylight in Chicago, IL, USA. University of Chicago's Animal facility was maintained at 25 degree c and 30-70% humidity. |
| Wild animals | The study did not involve wild animals. |
| Reporting on sex | Only female mice were used for experiments. |
| Field-collected samples | The study did not involve samples collected from the field. |

| Ethics oversight | All experiments were performed in accordance with the US National Institutes of Health Guide for the Care and Use of Laboratory Animals and approved by The University of Chicago Institutional Animal Care and Use Committee. |

Note that full information on the approval of the study protocol must also be provided in the manuscript.

# Flow Cytometry

## Plots

Confirm that:

☒ The axis labels state the marker and fluorochrome used (e.g. CD4-FITC).

☒ The axis scales are clearly visible. Include numbers along axes only for bottom left plot of group (a 'group' is an analysis of identical markers).

☒ All plots are contour plots with outliers or pseudocolor plots.

☒ A numerical value for number of cells or percentage (with statistics) is provided.

## Methodology

| Sample preparation | To analyze splenic B cells, total splenocytes were obtained by mashing spleens on 70-μm filters followed by red blood cell lysis (Lonza). To analysis red blood cell content in the bone marrow, total bone marrow cells were flushed out of femora and tibiae using PBS. Single-cell suspensions were stained in the presence of Fc receptor-blocking antibodies (anti-mouse CD16/32, clone 93) using the following antibodies (BioLegend): CD19-FITC (clone 1D3/CD19, 152403), B220-PerCP (clone RA3-6B2, 103233), CD93-PE (clone AA4.1, 136503), CD23-APC (clone B3B4, 101619), CD21-Pacific Blue (clone 7E9, 123413), Ter119-FITC (clone TER-119, 116205), CD45-APC-Cy7 (clone 30-F11, 103115). Cell viability was measured using Zombie Yellow Fixable Viability kit (423103) or DAPI. |
| Instrument | Flow Cytometry analysis was performed on a NovoCyte flow cytometer (Acea Biosciences/Agilent) |
| Software | Flow cytometry data were analyzed with FlowJo (Version 10.6.2) software. |
| Cell population abundance | Purity was assessed by Flow cytometry analysis. Absolute numbers of cells are outlined in relevant Figures. |
| Gating strategy | Events were initially gated by FSC-A and SSC-A and then FSC-A and FSC-H were used to exclude doublets. Live cells were gated using a viability dye. Subsequent gating depends on the population of interest and is outlined in Supplementary Information. |

☒ Tick this box to confirm that a figure exemplifying the gating strategy is provided in the Supplementary Information.

