## [Peer Review File · Nature Immunology]

Peer Review Information

Journal: Nature Immunology

Manuscript Title: A Pairwise Cytokine Code Explains the Organism-Wide Response to Sepsis

Corresponding author name(s): Professor Nicolas Chevrier

Reviewer Comments & Decisions:

Decision Letter, initial version:
--

8th Jun 2023

Dear Nicolas,

Thank you for providing a point-by-point response to the referees comments on your manuscript entitled, "A Pairwise Cytokine Code Explains the Organism-Wide Response to Sepsis". As noted in my previous message, while they find your work of considerable potential interest, they have raised quite substantial concerns that must be addressed. In light of these comments, we cannot accept the current manuscript for publication, but would be very interested in considering a revised version along the lines proposed in your rebuttal.

We invite you to submit a substantially revised manuscript, however please bear in mind that we will be reluctant to approach the referees again in the absence of major revisions.

Specifically, the revision should include new experiments to address:

(1) add data comparing LPS challenge to three grades of CLP severity (mild, moderate & severe by varying the position of cecal ligation and number/size of the cecal punctures) at 6h and 24 h time points.

(2) profile changes in gene expression across the tissues upon CLP in the presence or absence of cytokine blockade.

Additionally, referee #3 noted in their comments to editors:

"I think it should be a terrific resource. The obvious thing that occurs to me is how accessible the data will be for readers - I wonder if the current ms would provide a similar platform for modestly skillful investigators to use the impressive datasets."

Please include the additional textual clarifications as indicated in your response letter.

When you revise your manuscript, please take into account all reviewer and editor comments, please highlight all changes in the manuscript text file in Microsoft Word format.

* If you have not done so already please begin to revise your manuscript so that it conforms to our Article format instructions at <http://www.nature.com/ni/authors/index.html>. Refer also to any guidelines provided in this letter.

The Reporting Summary can be found here:

When submitting the revised version of your manuscript, please pay close attention to our [href="https://www.nature.com/nature-portfolio/editorial-policies/image-integrity">Digital Image Integrity Guidelines.](https://www.nature.com/nature-portfolio/editorial-policies/image-integrity) and to the following points below:

[REDACTED]

If you wish to submit a suitably revised manuscript we would hope to receive it within 6 months. If you cannot send it within this time, please let us know. We will be happy to consider your revision so long as nothing similar has been accepted for publication at Nature Immunology or published

elsewhere.

Nature Immunology is committed to improving transparency in authorship. As part of our efforts in this direction, we are now requesting that all authors identified as 'corresponding author' on published papers create and link their Open Researcher and Contributor Identifier (ORCID) with their account on the Manuscript Tracking System (MTS), prior to acceptance. ORCID helps the scientific community achieve unambiguous attribution of all scholarly contributions. You can create and link your ORCID from the home page of the MTS by clicking on 'Modify my Springer Nature account'. For more information please visit www.springernature.com/orcid.

Thank you for the opportunity to review your work.

Kind regards,

Laurie

Laurie A. Dempsey, Ph.D.
Senior Editor
Nature Immunology
l.dempsey@us.nature.com
ORCID: 0000-0002-3304-796X

Referee expertise:

Referee #1: Sepsis

Referee #2: Inflammation/sepsis/trauma

Referee #3: Cytokine signaling/transcriptomics

Reviewers' Comments:

Reviewer #1:

Remarks to the Author:

The manuscript by Takahama et al attempts to 'decode the chaos' in systemic cytokine signaling in the early phases of (severe) sepsis into simplifying handpicked pairwise cytokine message and provides spatiotemporal data key to start (potentially) compiling a mechanistic framework for the impact of sepsis on the host.

Sepsis is defined as a life-threatening organ dysfunction caused by dysregulated host response to infection and pathogenesis of sepsis is indeed poorly defined. The questions is why we are having so much trouble understanding sepsis? Sepsis as a syndrome is highly complex and is influenced by a)

pathogen (type, load, and virulence); b) host (genetics, epigenetics, comorbidities); c) environment (history of pathogen encounters, microbiome...); and d) time elapsed after the start of the infection with distinct response at local, regional, and systemic levels. Therefore, 'decoding the sepsis-induced chaos' is highly significant. In addition, this submission has tremendous amount of data that, once publicly available, could help/guide/instruct more in-depth analyses to aid in our understanding of sepsis pathogenesis.

However significant issues were noted that influence overall assessment of this manuscript. For instance:

- a) Model used to interrogate sepsis. How LPS injections truly recapitulates sepsis? Second model used (CLP) is likely more relevant but seldom used and since all mice died in the first 48 hours after surgery one could argue that this models just subset of patients and that relationship observed (results, conclusions drawn...) could and likely differ in host with less severe (and way more prominent) sepsis. This represents significant limitations of the study and should be discussed.
- b) This paper is an example of the disproportional amount of data presented compared to the amount of time/space/confirmatory experiments provided to support conclusions drawn.
- c) The notion that cytokine storm influence the pathogenesis of sepsis is not new. The notion that single cytokine controls the infection is dismissed as highly unlikely by many experiments and groups suggesting that multiple cytokines (and chemokines...) contribute to the sepsis development and organ-specific dysfunction. Of note (please see point a) – contribution of particular cytokines and interactions among them depend on the sepsis severity, model used, status of the host...etc... In addition, one could argue that conclusions drawn here have some conceptual similarities to the paper (PMID: 33278357) where the authors identify the pair of TNF-a and IFN-g (one of the pairs this submission also explore) to play a role in causing inflammatory cell death, tissue damage, and mortality in SARS-COV2, and cytokine shock syndromes but the NI submission does not cite them. For instance, the similarities include 1) This paper shows the survival of LPS-treated mice is dramatically improved upon blockade of both TNF-a and IFN-g (similar to 3E in the NI submission) and 2) this paper shows increased ALT, BUN levels upon TNF-a+IFN-g treatment (similar to Fig 3G in the NI submission).

Reviewer #2:

Remarks to the Author:

Review of NI-A35727 for Nature Immunology

In the submitted manuscript titled "A Pairwise Cytokine Code Explains the Organism-Wide Response to Sepsis" the authors with highly acknowledged research expertise in dynamic mapping of immune processes at the organism level provide a comprehensive, descriptive experimental study addressing spatio-temporal gene expression across organs during systemic inflammation.

The authors build upon their previous findings, which suggest that "processing of multiple input signals by innate immune cells is governed by pairwise effects" (Surya Pandey et al., Cell Systems, 2020). In this study, they conduct comprehensive and well-executed experiments to investigate the effects of exposure to either single or pairwise recombinant cytokines, as well as LPS injection or CLP procedure, without any specific hypothesis. The authors employ state-of-the-art technology for the primary measurements. The main conclusion drawn by the authors is that they have unraveled the intricate cytokine interplay ("cacophony") in sepsis, revealing a pairwise cytokine communication at the gene, cell, and tissue levels. They suggest that three cytokine pairs can effectively replicate the

majority of transcriptional, physiological, and fitness responses observed in sepsis. However, caution must be exercised when drawing conclusions in the context of sepsis, as the findings may be premature, and the translatability to the clinical setting remains uncertain. Nonetheless, this topic and the original findings align perfectly with the scope of Nature Immunology and is expected to generate significant interest among the general readership.

There is one major and some minor concerns as described herewith:
Major concerns:

1. The title of the study posits that "a pairwise cytokine code explains the organism-wide response to sepsis". While the provocative findings hold true for a systemic inflammatory response in general, their applicability to sepsis remains inconclusive due to the reliance on models primarily based on data generated from the LPS-model. It is widely recognized that LPS injection does not adequately replicate the clinical features and course of sepsis, instead representing an acute LPS endotoxemia (referred to as LPS "intoxication"), characterized by transient robust inflammatory response features. In fact, an international expert consensus on preclinical sepsis studies emphasizes that LPS administration should not be regarded as a relevant model of sepsis, given the substantial disparities between the sepsis phenotype and that induced by LPS (reviewed in Osuchowski et al., Shock 2018; Libert et al., Shock 2019). Additionally, it is noteworthy that while anti-TNF treatment demonstrated efficacy in the LPS mouse model, it did not exhibit the same benefit in the CLP mouse model (Remick et al., Shock 1995). In the present study, the authors utilized the CLP-model and compared it to LPS-exposure. However, the selected time window of 12 hours following LPS challenge and 16 hours following CLP does not rule out the possibility that the early transient cytokine response (e.g., IL-1beta, IL-6, IFN-gamma, as shown in Fig. 2B) triggered by LPS exposure merely coincided with the slowly escalating response observed in the case of CLP (as depicted in the attached scheme). Can the authors confidently assert that within the first 6 hours (early phase) or around 24 hours (when CLP-sepsis typically presents clinically in full manifestation) and beyond, both models still exhibit a comparable organism-wide response? Furthermore, it is noteworthy that the overlap in the identity of LPS and CLP exposures is significant but varies, ranging from 29% (heart) to less than 50% (colon, iLN, spleen, BM), with a maximum of 68% (thymus), indicating still substantial differences. Unless experimentally demonstrated otherwise, statements regarding similar responses in the context of "sepsis" should be approached with caution and instead employ the term "endotoxemia" throughout the whole manuscript to better reflect the observed outcomes.

Minor concerns:

1. Title: The title should indicate the exploratory nature of the study and refer to endotoxemia rather than to "sepsis" (see above).
2. Results: After LPS-exposure (sub-lethal dose), the body temperature drops in one case from ca. 38°C to 27°C (Fig. 3F) and in the other case from ca. 38°C to 32°C (Fig. 3H) . Are there any explanation for the difference?
3. Results: Fig. 3F: the y-axes should display "body temperature (°C)"
4. Results: Fig. 3E: Percent survival with such a small n-size is inadequate – could the authors display the individual decease (e.g. in a Kaplan-Meier-Curve or just in a table); or did e.g. all control animals die at 50 h after exposure?
5. Results: Is the body temperature the only real physiological parameters or were others also measured and correlated to the spatio-temporal genetic response?
6. Discussion (pg. 16): "...we found, that non-lymphoid tissue regained homeostasis sooner than lymphoid ones..." again might only true for a LPS-endotoxemia model but not for a CLP-sepsis model

where homeostasis is more difficult (or never) to achieve.

pg. 17: "most of these cellular effects were previously observed in sepsis,..." – reference 40 is not referring to sepsis but rather to an LPS-model.

7. In the preprint (bioRxiv. 2023 Feb 2:2023.01.30.526342. doi: 10.1101/2023.01.30.526342) the authors also mentioned that the secreted phospholipase PLA2G5 mediates hemolysis in blood, contributing to organ failure during sepsis. Have the authors integrated and displayed those data as well?

Reviewer #3:

Remarks to the Author:

In this work, the authors aimed to capture comprehensive spatiotemporal effects of sepsis (LPS, CLP) on a whole body scale (13 tissues) via bulk mRNA-seq (Fig.1). Subsequently, they endeavored to mimic sepsis by in vivo injection of recombinant cytokines. They tested 6 cytokines separately and in combination (15 cytokine pairs) to identify 3 cytokine pairs (all containing TNF) as regulating key genes (DEGs) most similar to the bona fide sepsis profile (Fig.2). Furthermore, they performed computational analysis (topic modeling) to infer cell type abundance in organs from bulk mRNA-seq data and identified 195 cell types in 9 tissues in 9 experimental perturbations. They found lymphopenia, thymic involution, and endothelial cell expansion in relevant organs at relevant times during sepsis (Fig.4). Based on those observations, they performed validation experiments. First, they did genetic and pharmacologic blockade of key cytokines (TNF, IL1b,IFNg, IL18) to ameliorate the sepsis phenotype (Fig.3), confirming the significant involvement of pertinent cytokines. Second, they performed whole body spatial transcriptomic analysis to confirm negative impact over select cell types in select organs (Fig.5), as well as mobilization of neutrophils into tissues and depletion of B cell subsets from spleen (Fig.6).

This is a tour de force paper with large scale sampling to understand dynamic process of sepsis response in a comprehensive manner. Because of the ambitious scale of the work, not all aspect of their observations are explained by a simple principle (the pairwise cytokine code), but rather is left for future follow up. Nonetheless, the work offers a significant resource to understand sepsis better with a unique perspective of whole body view.

The following are specific issues that the authors should consider:

Fig.2C

- positive control sepsis plasma produced less than 30% of overlap with sepsis genes. Isn't it too low? Is it because of "dose" of plasma?
- Nonlinear effect is not clearly defined or explained. Is it equal to non-additive, i.e., both synergistic and antagonistic combined?

Fig.2F X-axis

- what does -70% overlap mean? 70% overlap in negatively regulated genes?
- " a large fraction of overlap" - the statement is probably true for rTNF+rIL18 but not for other conditions. It is a bit misleading to describe less than 50% of overlap as a large fraction.

Fig.4 Cell type abundance inference based on cell type marker gene expression.

- How accurate would it be? What if per cell gene expression changes upon stimulation in addition to

increase in cell number? Would it overestimate cell abundance? Any thoughts on that aspect?

Fig. S5B – there is no S5B, perhaps the label is missing. It is also not clear what kind of criteria were used to divide mostly share vs mostly pair specific response. Just an overall impression?

Fig.S9 – lung is an exception not explained by the paired cytokine code. Any in depth consideration what the mechanism might be?

Author Rebuttal to Initial comments

See inserted PDF

Response to Reviewers' Comments

We thank the reviewers for their careful reading and insightful comments about our manuscript. We are pleased that the reviewers are highly enthusiastic about the novelty and impact of this study, which they refer to as “*highly significant*” (Reviewer 1), “*comprehensive and well executed*” (Reviewer 2), and a “*tour de force paper [...] offering a significant resource to understand sepsis better with a unique perspective of whole body view*” (Reviewer 3). Furthermore, Reviewer 2 mentions that “*this topic and the original findings align perfectly with the scope of Nature Immunology and is expected to generate significant interest among the general readership.*”

In the revised manuscript, we have addressed all the reviewers' suggestions, concerning the text, experiments, and analyses. We first highlight the key revisions and additions, and then provide a point-by-point response, detailing these and all the additional experiments, analyses, and revisions to the manuscript.

The key experimental additions and revisions to this manuscript include:

- **We compare the effects of LPS and CLP on gene expression across 8 organ types at three time points: 0.25-, 0.5-, and 1-day after LPS injection or mild, moderate, or severe CLP (Fig. 2 and Extended Data Fig. 2, Methods).** We found a high degree of similarity in the organism-wide expression profiles induced by both disease models. Some of the observed differences are likely due to the presence of bacterial dissemination in CLP but not in LPS, although this would need testing in future work.
- **We measure the organism-wide effects of pairwise cytokine perturbations on tissue states during CLP using TNF neutralizing antibodies in *Il1b*^{-/-}, *Il18*^{-/-}, or *Ifng*^{-/-} mice (Fig. 4c,g and Extended Data Fig. 6).** We found that pairwise cytokine perturbations, but not TNF blockade alone, by and large rescued the effects of severe CLP on tissue responses. Thus, these results further support our proposed model on the central role of TNF plus IL-1 β , IL-18, or IFN- γ in the organism-wide response of the host to sepsis.

Reviewer #1

(Remarks to the Author)

The manuscript by Takahama et al attempts to 'decode the chaos' in systemic cytokine signaling in the early phases of (severe) sepsis into simplifying handpicked pairwise cytokine message and provides spatiotemporal data key to start (potentially) compiling a mechanistic framework for the impact of sepsis on the host.

Sepsis is defined as a life-threatening organ dysfunction caused by dysregulated host response to infection and pathogenesis of sepsis is indeed poorly defined. The questions is why we are having so much trouble understanding sepsis? Sepsis as a syndrome is highly complex and is influenced by a) pathogen (type, load, and virulence); b) host (genetics, epigenetics, comorbidities); c) environment (history of pathogen encounters, microbiome...); and d) time elapsed after the start of the infection with distinct response at local, regional, and systemic levels. Therefore, 'decoding the sepsis-induced chaos' is highly significant. In addition, this submission has tremendous amount of data that, once publicly available, could help/guide/instruct more in-depth analyses to aid in our understanding of sepsis pathogenesis.

However significant issues were noted that influence overall assessment of this manuscript. For instance:

a) Model used to interrogate sepsis. How LPS injections truly recapitulates sepsis? Second model used (CLP) is likely more relevant but seldom used and since all mice died in the first 48 hours after surgery one could argue that this models just subset of patients and that relationship observed (results, conclusions drawn...) could and likely differ in host with less severe (and way more prominent) sepsis. This represents significant limitations of the study and should be discussed.

We thank the reviewer for raising this important point. We agree with Reviewer 1 that CLP is widely thought to be a better representation of human sepsis. We used CLP in our initial submission and noted the following about CLP in the Results section (p. 7): "Next, we compared the organism-wide effects of LPS to those obtained with three different severity grades of CLP (Extended Data Fig. 2a,b), a polymicrobial infection starting in the abdominal cavity which is considered the gold standard model for sepsis due to its high clinical relevance^{22,23}."

In this revised manuscript, we added new experiments in which we used CLP procedures that model three levels of disease severity (mild, moderate, severe) by varying the position of the cecal ligation and the number and size of cecal punctures (Extended Data Fig. 2a,b and Methods), by following procedures reported by others (see Ref. 22 in our manuscript). Results from these experiments are presented in Fig. 2 and Extended Data Fig. 2 and described in detail in the Results section (see page 7 bottom paragraph and page 8 top paragraph).

Overall, our new data suggest a high degree of similarity between the whole-tissue gene expression profiles of LPS and mild, moderate, and severe CLP at 0.25-, 0.5-, and 1-day post-sepsis induction. The overlap between LPS and CLP was the highest for severe CLP, which led to more differential expression than mild and moderate CLP sepsis, as could be anticipated based on the body temperature and survival (Extended Data Fig. 2a,b). Notably, while some changes in gene expression were not found to be statistically significant, the trend in log-fold changes were often similar across all tested conditions (LPS, CLP) and times (Fig. 2a-c and Extended Data Fig. 2c). This discrepancy could be due to the noisier nature of CLP surgeries compared to simply injecting a bolus of LPS intra-peritoneally, leading to more statistical uncertainty in RNA-seq.

b) This paper is an example of the disproportional amount of data presented compared to the amount of time/space/confirmatory experiments provided to support conclusions drawn.

We thank the reviewer for raising this point, which we see as a unique opportunity for future work to transform our understanding of sepsis and cytokine storms as opposed to a weakness of the present study. Our work provides the first look, to our knowledge, at the pathogenesis of sepsis and cytokine storms from the perspective of the whole body. While answering this question inherently requires generating a significant amount of data, we thoroughly validated our proposed model on the pairwise cytokine code of sepsis (Fig. 3) using perturbations (KO, blockade) followed by molecular, cellular, and physiological readouts (Fig. 4 and Extended Data Fig. 6). Furthermore, we validated the predicted cellular effects of sepsis and cytokine pairs on 195 cell types in 9 tissues (Fig. 5 and Extended Data Fig. 7) in 7 case studies: liver and kidney epithelia, colon neurons, red blood cells, multi-tissue macrophages and neutrophils, and splenic B cells (Fig. 6-7 and Extended Data Fig. 8-10). Thus, our results collectively provide a mechanistic basis for multiple known and previously unreported cellular effects of sepsis across multiple tissues. Moreover, the organism-wide datasets reported in our work will lay the foundation for numerous follow-up mechanistic studies and new avenues for sepsis therapy research.

c) The notion that cytokine storm influence the pathogenesis of sepsis is not new. The notion that single cytokine controls the infection is dismissed as highly unlikely by many experiments and groups suggesting that multiple cytokines (and chemokines...) contribute to the sepsis development and organ-specific dysfunction. Of note (please see point a) – contribution of particular cytokines and interactions among them depend on the sepsis severity, model used, status of the host...etc... In addition, one could argue that conclusions drawn here have some conceptual similarities to the paper (PMID: 33278357) where the authors identify the pair of TNF- α and IFN- γ (one of the pairs this submission also explore) to play a role in causing inflammatory cell death, tissue damage, and mortality in SARS-COV2, and cytokine shock syndromes but the NI submission does not cite them. For instance, the similarities include 1) This paper shows the survival of LPS-treated mice is dramatically improved upon blockade of both TNF- α and IFN- γ (similar to 3E in the NI submission) and 2) this paper shows increased ALT, BUN levels upon TNF- α +IFN- γ treatment (similar to Fig 3G in the NI submission).

We thank the reviewer for raising these points about prior knowledge on cytokines in sepsis.

First, we agree with the assessment that while the importance of systemic cytokine signaling in sepsis has been appreciated for decades, we still lack a unifying framework to explain how these cytokines collectively impact the body, as noted in paragraph 2 of our Introduction on page 3 (bottom) and 4 (top): “However, the links between the molecular and cellular factors that produce the damaging impact of sepsis for the body have not been systematically mapped. For example, the uncontrolled, systemic activity of cytokines contributes to tissue injury and organ failure¹³, but it is unclear which cytokines – alone or in combination – impact which cells and tissues across the body. This gap in knowledge is due to features of the cytokine language which make it hard to decode, such as the variations in concentrations (local and systemic), activities (pro, anti, or both for any given cytokine), and interactions within a mixture of cytokines present in a tissue. As a result, we lack a unifying framework to understand how the cytokine network functions in sepsis, including the network’s target cells, hierarchy, interactions, and feedback loops¹⁴.”

Second, we did cite the work by Karki and colleagues (PMID 33278357, ref. 53 in our initial and revised manuscripts), as well as works from other groups on the effects of TNF and IFN- γ (see references listed below), in the discussion section's following part (page 19): "For example, TNF has been shown to combine synergistically or antagonistically with IFN- γ or IL-1 β to impact secretion, cell death or proliferation, and cell states in immune and non-immune cells in culture⁴¹⁻⁵⁰. While the interaction between TNF and IL-18 had not been reported to our knowledge, TNF plus IFN- γ ⁵¹⁻⁵⁴ or IL-1 β ⁵⁵⁻⁵⁷ worsen the outcome of sepsis and other inflammatory disorders *in vivo*. The cytokines of this module also influence each other's production¹³, which further supports the hierarchy uncovered by our pairwise cytokine screening data."

Lastly, we would like to clarify why our work differs in several ways from the works of Karki and colleagues and others before them, as referenced in the paragraph above, on the effects of TNF plus IFN- γ . First, we measured the effects of 15 cytokine pairs on tissue states across the body, including quantitative analyses of nonlinear effects on each gene regulated in each tissue. Second, we identified the target cell types of the three key cytokine pairs from our work out of 195 cell types sampled across 9 tissues. Third, our data reveals a simplifying rule in cytokine storms, whereby the effects of three cytokine pairs suffice to explain the collective behavior of multiple cytokines on tissues across the body. On the contrary, previous works as those cited above focused on the effects of a single cytokine pair at a time (*e.g.*, TNF plus IFN- γ or IL-1 β) on a given biological process (*e.g.*, survival, cell death), which yielded critically important but fundamentally distinct knowledge from our study, which relates to the broader context of how the systemic cytokine network as a whole impacts the whole body of the host during uncontrolled inflammation such as sepsis. In other words, prior work on the roles of cytokines in sepsis did not address how the cytokine network functions collectively impact tissues of the body. Conceptually, our results thus provide a new way of thinking about cytokine storm syndromes in general, which is likely to spur many avenues of investigations in the future.

Reviewer #2

(Remarks to the Author)

Review of NI-A35727 for Nature Immunology

In the submitted manuscript titled “A Pairwise Cytokine Code Explains the Organism-Wide Response to Sepsis” the authors with highly acknowledged research expertise in dynamic mapping of immune processes at the organism level provide a comprehensive, descriptive experimental study addressing spatio-temporal gene expression across organs during systemic inflammation. The authors build upon their previous findings, which suggest that “processing of multiple input signals by innate immune cells is governed by pairwise effects” (Surya Pandey et al., Cell Systems, 2020). In this study, they conduct comprehensive and well-executed experiments to investigate the effects of exposure to either single or pairwise recombinant cytokines, as well as LPS injection or CLP procedure, without any specific hypothesis. The authors employ state-of-the-art technology for the primary measurements. The main conclusion drawn by the authors is that they have unraveled the intricate cytokine interplay (“cacophony”) in sepsis, revealing a pairwise cytokine communication at the gene, cell, and tissue levels. They suggest that three cytokine pairs can effectively replicate the majority of transcriptional, physiological, and fitness responses observed in sepsis. However, caution must be exercised when drawing conclusions in the context of sepsis, as the findings may be premature, and the translatability to the clinical setting remains uncertain. Nonetheless, this topic and the original findings align perfectly with the scope of Nature Immunology and is expected to generate significant interest among the general readership. There is one major and some minor concerns as described herewith:

Major concerns:

1. The title of the study posits that “a pairwise cytokine code explains the organism-wide response to sepsis”. While the provocative findings hold true for a systemic inflammatory response in general, their applicability to sepsis remains inconclusive due to the reliance on models primarily based on data generated from the LPS-model. It is widely recognized that LPS injection does not adequately replicate the clinical features and course of sepsis, instead representing an acute LPS endotoxemia (referred to as LPS “intoxication”), characterized by transient robust inflammatory response features. In fact, an international expert consensus on preclinical sepsis studies emphasizes that LPS administration should not be regarded as a relevant model of sepsis, given the substantial disparities between the sepsis phenotype and that induced by LPS (reviewed in Osuchowski et al., Shock 2018; Libert et al., Shock 2019). Additionally, it is noteworthy that while anti-TNF treatment demonstrated efficacy in the LPS mouse model, it did not exhibit the same benefit in the CLP mouse model (Remick et al., Shock 1995).

In the present study, the authors utilized the CLP-model and compared it to LPS-exposure. However, the selected time window of 12 hours following LPS challenge and 16 hours following CLP does not rule out the possibility that the early transient cytokine response (e.g., IL-1beta, IL-6, IFN-gamma, as shown in Fig. 2B) triggered by LPS exposure merely coincided with the slowly escalating response observed in the case of CLP (as depicted in the attached scheme). Can the authors confidently assert that within the first 6 hours (early phase) or around 24 hours (when CLP-sepsis typically presents clinically in full manifestation) and beyond, both models still exhibit a comparable organism-wide response? Furthermore, it is noteworthy that the overlap in the identity of LPS and CLP exposures is significant but varies, ranging from 29% (heart) to less than

50% (colon, iLN, spleen, BM), with a maximum of 68% (thymus), indicating still substantial differences. Unless experimentally demonstrated otherwise, statements regarding similar responses in the context of "sepsis" should be approached with caution and instead employ the term "endotoxemia" throughout the whole manuscript to better reflect the observed outcomes.

We thank Reviewer 2 for raising these points about the CLP model. In our initial manuscript, we had only compared LPS to severe CLP at 0.5-day post-LPS injection or CLP surgery (original Fig. S2). In this revised manuscript, we added new experiments in which we used CLP procedures that model three levels of disease severity (mild, moderate, severe) by varying the position of the cecal ligation and the number and size of cecal punctures (Extended Data Fig. 2a,b and Methods), by following procedures reported by others (see Ref. 22 in our manuscript). Results from these experiments are presented in Fig. 2 and Extended Data Fig. 2 and described in detail in the Results section (see page 7 bottom paragraph and page 8 top paragraph). Overall, our new data suggest a high degree of similarity between the whole-tissue gene expression profiles of LPS and mild, moderate, and severe CLP at 0.25-, 0.5-, and 1-day post-sepsis induction. The overlap between LPS and CLP was the highest for severe CLP, which led to more differential expression than mild and moderate CLP sepsis, as could be anticipated based on the body temperature and survival (Extended Data Fig. 2a,b). Notably, while some changes in gene expression were not found to be statistically significant, the trend in log-fold changes were often similar across all tested conditions (LPS, CLP) and times (Fig. 2a-c and Extended Data Fig. 2c). This discrepancy could be due to the noisier nature of CLP surgeries compared to simply injecting a bolus of LPS intra-peritoneally, leading to more statistical uncertainty in RNA-seq.

Going further, we added new experiments in which we profiled changes in gene expression across the tissues upon CLP in the presence or absence of cytokine blockade (*i.e.*, anti-TNF +/- genetic deletions of IL-1 β , IL-18, or IFN- γ) (see revised text on page 11) (Fig. 4, Extended Data Fig. 6, and Supplementary Table 5). We found that all three pairwise cytokine perturbations counteracted most of the gene expression changes due to CLP sepsis, regardless of the severity grade of CLP. Interestingly, TNF neutralization alone induced little to no statistically significant changes in tissue expression during CLP, although, in log-fold change space, we observed that many genes showed a trend in expression which was opposite to that of CLP effects without cytokine neutralization (Extended Data Fig. 6c and Supplementary Table 5c,d).

These results on the impact of anti-TNF during CLP sepsis are linked to the point raised by Reviewer 2 on the work of Remick and colleagues (Remick et al., Shock, 1995). Several technical considerations from the paper by Remick and colleagues are worth noting: the genetic background of the mice were BALB/c and CD-1, not C57BL6/J as in our study. BALB/c mice are known to be more resistant to sepsis. In addition, Remick and colleagues used anti-TNF anti-serum, which is likely to perform differently from the monoclonal antibodies used in our work. Lastly, the effects of blocking TNF during CLP on tissue states across the body had never been examined, to our knowledge, before the new experiments described above.

Lastly, we added the term "endotoxemia" in figures, legends, and text of this revised manuscript to address the reviewer's comment and improve clarity.

Minor concerns:

1. Title: The title should indicate the exploratory nature of the study and refer to endotoxemia rather than to “sepsis” (see above).

The new data described above on CLP across severity grades and cytokine perturbations address this point.

2. Results: After LPS-exposure (sub-lethal dose), the body temperature drops in one case from ca. 38°C to 27°C (Fig. 3F) and in the other case from ca. 38°C to 32°C (Fig. 3H) . Are there any explanation for the difference?

These differences are explained by the fact that we used two different doses of LPS in these two panels: sublethal (3F in initial submission, Fig. 4f in revised figures) and lethal (3H in initial, 4i in revised). We apologize for an error in the legend of Fig. 3F, which should have read “lethal” not “sublethal” dose of LPS. We corrected the legend of revised Fig. 4.

3. Results: Fig. 3F: the y-axis should display “body temperature (°C)”

We corrected this error by labeling the Y axis in revised Fig. 4f (formerly Fig. 3F).

4. Results: Fig. 3E: Percent survival with such a small n-size is inadequate – could the authors display the individual decease (e.g. in a Kaplan-Meier-Curve or just in a table); or did e.g. all control animals die at 50 h after exposure?

All control animals died at 48 hours after exposure, and we displayed the individual decease in revised Fig. 4e (formerly Fig. 3E).

5. Results: Is the body temperature the only real physiological parameters or were others also measured and correlated to the spatio-temporal genetic response?

While we are not certain about which figure is being discussed in this point, we did measure changes in body temperature, plasma levels of tissue injury markers, and survival in revised Fig. 4 (formerly Fig. 3).

6. Discussion (pg. 16): “...we found, that non-lymphoid tissue regained homeostasis sooner than lymphoid ones...” again might only true for a LPS-endotoxemia model but not for a CLP-sepsis model where homeostasis is more difficult (or never) to achieve.

We clarified this paragraph as follows, with underlined parts reflecting newly added text: “Second, we found that non-lymphoid tissues regained homeostasis sooner than lymphoid ones in endotoxemia. This result is reminiscent of how some organs reverse dysfunction in sepsis, including those poor at regenerating such as heart, lung, kidney, or brain, whereas the immune system suffers long-term dysregulation with life-threatening consequences for survivors^{20,26}. Further mining our data might help to identify factors that safeguard non-lymphoid tissues, such as IL-10 for microglia³⁷ or GDF15 for heart³⁸ which are both present in our data, or those that damage lymphoid tissues and cells, such as pairwise effects in the cytokine network. Future work

is needed to assess the dynamics of tissue recovery, if any, during CLP sepsis with or without antibiotics treatment mimicking human patient treatment regimens.”

pg. 17: “most of these cellular effects were previously observed in sepsis, ...” – reference 40 is not referring to sepsis but rather to an LPS-model.

We revised this sentence as follows (page 18): “Most of these cellular effects were previously observed in sepsis or endotoxemia, such as an increase in thymic macrophages³⁹, erythropenia²⁸, splenic B cell loss¹⁵, or changes in kidney tubules⁴⁰ but, crucially, lacked causal factors.”

7. In the preprint (bioRxiv. 2023 Feb 2:2023.01.30.526342. doi: 10.1101/2023.01.30.526342) the authors also mentioned that the secreted phospholipase PLA2G5 mediates hemolysis in blood, contributing to organ failure during sepsis. Have the authors integrated and displayed those data as well?

We thank the reviewer for also reading our pre-print from last January. Our findings on PLA2G5 are part of a separate, dedicated manuscript unrelated to this submission. We initially posted our findings on the cytokine code and PLA2G5 in sepsis together in a single pre-print for grant submission purposes at the time. Significant new data has been generated for both studies compared to our pre-print, as shown in this submission on the cytokine code discovery.

Reviewer #3

(Remarks to the Author)

In this work, the authors aimed to capture comprehensive spatiotemporal effects of sepsis (LPS, CLP) on a whole body scale (13 tissues) via bulk mRNA-seq (Fig.1). Subsequently, they endeavored to mimic sepsis by in vivo injection of recombinant cytokines. They tested 6 cytokines separately and in combination (15 cytokine pairs) to identify 3 cytokine pairs (all containing TNF) as regulating key genes (DEGs) most similar to the bona fide sepsis profile (Fig.2). Furthermore, they performed computational analysis (topic modeling) to infer cell type abundance in organs from bulk mRNA-seq data and identified 195 cell types in 9 tissues in 9 experimental perturbations. They found lymphopenia, thymic involution, and endothelial cell expansion in relevant organs at relevant times during sepsis (Fig.4). Based on those observations, they performed validation experiments. First, they did genetic and pharmacologic blockade of key cytokines (TNF, IL1b,IFNg, IL18) to ameliorate the sepsis phenotype (Fig.3), confirming the significant involvement of pertinent cytokines. Second, they performed whole body spatial transcriptomic analysis to confirm negative impact over select cell types in select organs (Fig.5), as well as mobilization of neutrophils into tissues and depletion of B cell subsets from spleen (Fig.6). This is a tour de force paper with large scale sampling to understand dynamic process of sepsis response in a comprehensive manner. Because of the ambitious scale of the work, not all aspect of their observations are explained by a simple principle (the pairwise cytokine code), but rather is left for future follow up. Nonetheless, the work offers a significant resource to understand sepsis better with a unique perspective of whole body view.

The following are specific issues that the authors should consider:

Fig.2C

- positive control sepsis plasma produced less than 30% of overlap with sepsis genes. Isn't it too low? Is it because of "dose" of plasma?

Our goal with using plasma transfer was to provide a qualitative rather than a quantitative benchmark, strictly speaking. We used 100 μ L of plasma (Methods) and agree with the reviewer that it is difficult to gauge the optimal amount to transfer. In addition to testing various volumes of plasma to transfer to naïve recipient mice, we would also need to test sampling plasma at different times post-LPS or CLP sepsis to better understand the effects of plasma factors present in different amounts depending on disease stage on tissues.

- Nonlinear effect is not clearly defined or explained. Is it equal to non-additive, i.e., both synergistic and antagonistic combined?

We apologize for the lack of clarity. The reviewer is correct: nonlinear effects refer to non-additive (synergy, antagonism) changes in gene expression induced by cytokine pairs relative to their matching singles. We clarified this in the revised Results section (last sentence of page 9). In addition, we have explained in detail the methods used to quantify nonlinear interaction at the level of gene expression changes in the Methods section entitled "Statistical modeling of cytokine pairwise effects on tissue gene expression" (see p. 56-58).

Fig.2F X-axis

- what does -70% overlap mean? 70% overlap in negatively regulated genes?

Yes, negative values indicate overlap in the identity of downregulated genes. We clarified the legends of this and all similar panels throughout the figures of our revised manuscript (e.g., Fig. 3f).

- “a large fraction of overlap” - the statement is probably true for rTNF+rIL18 but not for other conditions. It is a bit misleading to describe less than 50% of overlap as a large fraction.

We rephrased the sentence for clarity (page 10, bottom): “We found that, as in LPS, the tissue effects of TNF plus IL-18, IFN- γ , or IL-1 β collectively mirrored a large fraction of the effects of CLP and viral sepsis on tissues (**Fig. 3f**).”

In addition, we note that we used “large fraction” because while some changes in gene expression were not found to be statistically significant – and thus not counted in our overlap calculations – we found that the trends in log-fold changes were largely similar across the various conditions and time points used. These discrepancies are thus due to statistical thresholding as opposed to truly divergent biological profiles, as exemplified in the visual differences between Fig 3d vs 3f, or Extended Data Fig. 6a vs 6b.

Fig.4 Cell type abundance inference based on cell type marker gene expression. - How accurate would it be? What if per cell gene expression changes upon stimulation in addition to increase in cell number? Would it overestimate cell abundance? Any thoughts on that aspect?

We used two lines of evidence to increase our confidence in the accuracy of our predictions in terms of cell type calling and abundance. First, our computational predictions recapitulated most of the known cellular effects of sepsis. Second, we used experiments such as spatial transcriptomics, flow cytometry, or immunohistochemistry to validate half a dozen predictions. Future work will be needed to further validate predictions of interest using additional experiments.

Lastly, while formally answering this point would require a large-scale effort beyond the scope of this study which would require careful bulk and single-cell measurements, we tried to circumvent the issue by using gene sets which were the most specific for a cell type across many publicly available experiments (373 publications in total as mentioned in our Methods section), encompassing a vast array of steady and disease states. Therefore, we anticipate that these gene sets are likely to remain robust predictors of the relative abundance of a cell type regardless of the changes in the state of the cell.

Fig. S5B – there is no S5B, perhaps the label is missing. It is also not clear what kind of criteria were used to divide mostly share vs mostly pair specific response. Just an overall impression?

We fixed the text to mention S5A, not B (see Extended Data Fig. 5a in the revised manuscript).

Fig.S9 – lung is an exception not explained by the paired cytokine code. Any in depth consideration what the mechanism might be?

We agree with the reviewer that this is an interesting observation which would require further investigation to be confirmed and explained in terms of potential mechanism(s). Perhaps other factors are at play such as the coagulation and complement systems, although this is purely hypothetical as we have no data pointing to these or any other potential mechanisms.

Decision Letter, first revision:

9th Oct 2023

Dear Nicolas,

Thank you for submitting your revised manuscript "A Pairwise Cytokine Code Explains the Organism-Wide Response to Sepsis" (NI-A35727A). It has now been seen by the original referees and their comments are below. The reviewers find that the paper has improved in revision, and therefore we'll be happy in principle to publish it in Nature Immunology, pending minor revisions to satisfy the referees' final requests and to comply with our editorial and formatting guidelines.

We will now perform detailed checks on your paper and will send you a checklist detailing our editorial and formatting requirements in about a week. Please do not upload the final materials and make any revisions until you receive this additional information from us.

If you had not uploaded a Word file for the current version of the manuscript, we will need one before beginning the editing process; please email that to immunology@us.nature.com at your earliest convenience.

Thank you again for your interest in Nature Immunology Please do not hesitate to contact me if you have any questions.

Hopefully I will see you later this week at the upcoming joint retreat.

Kind regards,

Laurie

Laurie A. Dempsey, Ph.D.
Senior Editor
Nature Immunology
l.dempsey@us.nature.com
ORCID: 0000-0002-3304-796X

Reviewer #2 (Remarks to the Author):

The authors sufficiently addressed all the reviewers' concerns and thereby significantly improved the manuscript.

The additional key experiments on the CLP- versus LPS-induced systemic inflammatory reaction and the TNF-neutralizing antibodies in Il1b^{-/-}, IL18^{-/-}, Ifng^{-/-} mice are convincing and underline the overall findings.

Reviewer #3 (Remarks to the Author):

revised ms is acceptable

Final Decision Letter:

Dear Nicolas,

I am delighted to accept your manuscript entitled "A Pairwise Cytokine Code Explains the Organism-Wide Response to Sepsis" for publication in an upcoming issue of Nature Immunology.

Over the next few weeks, your paper will be copyedited to ensure that it conforms to Nature Immunology style. Once your paper is typeset, you will receive an email with a link to choose the appropriate publishing options for your paper and our Author Services team will be in touch regarding any additional information that may be required.

Please note that *Nature Immunology* is a Transformative Journal (TJ). Authors may publish their research with us through the traditional subscription access route or make their paper immediately open access through payment of an article-processing charge (APC). Authors will not be required to make a final decision about access to their article until it has been accepted. [Find out more about Transformative Journals](https://www.springernature.com/gp/open-research/transformative-journals).

Authors may need to take specific actions to achieve [compliance with funder and institutional open access mandates](https://www.springernature.com/gp/open-research/funding/policy-compliance-faqs). If your research is supported by a funder that requires immediate open access (e.g. according to [Plan S principles](https://www.springernature.com/gp/open-research/plan-s-compliance)) then you should select the gold OA route, and we will direct you to the compliant route where possible. For authors selecting the subscription publication route, the journal's standard licensing terms will need to be accepted, including [self-archiving policies](https://www.springernature.com/gp/open-research/policies/journal-policies). Those licensing terms will supersede

any other terms that the author or any third party may assert apply to any version of the manuscript.

Your paper will be published online soon after we receive your corrections and will appear in print in the next available issue. Content is published online weekly on Mondays and Thursdays, and the embargo is set at 16:00 London time (GMT)/11:00 am US Eastern time (EST) on the day of publication. Now is the time to inform your Public Relations or Press Office about your paper, as they might be interested in promoting its publication. This will allow them time to prepare an accurate and satisfactory press release. Include your manuscript tracking number (NI-A35727B) and the name of the journal, which they will need when they contact our office.

About one week before your paper is published online, we shall be distributing a press release to news organizations worldwide, which may very well include details of your work. We are happy for your institution or funding agency to prepare its own press release, but it must mention the embargo date and Nature Immunology. Our Press Office will contact you closer to the time of publication, but if you or your Press Office have any enquiries in the meantime, please contact press@nature.com.

Also, if you have any spectacular or outstanding figures or graphics associated with your manuscript - though not necessarily included with your submission - we'd be delighted to consider them as candidates for our cover. Simply send an electronic version (accompanied by a hard copy) to us with a possible cover caption enclosed.

If you have not already done so, we strongly recommend that you upload the step-by-step protocols used in this manuscript to the Protocol Exchange. Protocol Exchange is an open online resource that allows researchers to share their detailed experimental know-how. All uploaded protocols are made freely available, assigned DOIs for ease of citation and fully searchable through nature.com. Protocols can be linked to any publications in which they are used and will be linked to from your article. You can also establish a dedicated page to collect all your lab Protocols. By uploading your Protocols to Protocol Exchange, you are enabling researchers to more readily reproduce or adapt the methodology you use, as well as increasing the visibility of your protocols and papers. Upload your Protocols at www.nature.com/protocolexchange/. Further information can be found at www.nature.com/protocolexchange/about .

Please note that we encourage the authors to self-archive their manuscript (the accepted version

before copy editing) in their institutional repository, and in their funders' archives, six months after publication. Nature Portfolio recognizes the efforts of funding bodies to increase access of the research they fund, and strongly encourages authors to participate in such efforts. For information about our editorial policy, including license agreement and author copyright, please visit www.nature.com/ni/about/ed_policies/index.html

Kind regards,

Laurie

Laurie A. Dempsey, Ph.D.
Senior Editor
Nature Immunology
l.dempsey@us.nature.com
ORCID: 0000-0002-3304-796X